# Perforin-2 is essential for intracellular defense of parenchymal cells and phagocytes against pathogenic bacteria

Ryan M McCormack[1], Lesley R de Armas[1], Motoaki Shiratsuchi[1], Desiree G Fiorentino[1], Melissa L Olsson[1], Mathias G Lichtenheld[1], Alejo Morales[1], Kirill Lyapichev[1], Louis E Gonzalez[1], Natasa Strbo[1], Neelima Sukumar[2], Olivera Stojadinovic[3], Gregory V Plano[1], George P Munson[1], Marjana Tomic-Canic[3], Robert S Kirsner[3], David G Russell[2], Eckhard R Podack[1]*

[1]Department of Microbiology and Immunology, Miller School of Medicine, University of Miami, Miami, United States; [2]Department of Microbiology and Immunology, College of Veterinary Medicine, Cornell University, Ithaca, United States; [3]Wound Healing and Regenerative Medicine Research Program, Department of Dermatology and Cutaneous Surgery, Miller School of Medicine, University of Miami, Miami, United States

**Abstract** Perforin-2 (MPEG1) is a pore-forming, antibacterial protein with broad-spectrum activity. Perforin-2 is expressed constitutively in phagocytes and inducibly in parenchymal, tissue-forming cells. In vitro, Perforin-2 prevents the intracellular replication and proliferation of bacterial pathogens in these cells. Perforin-2 knockout mice are unable to control the systemic dissemination of methicillin-resistant *Staphylococcus aureus* (MRSA) or *Salmonella typhimurium* and perish shortly after epicutaneous or orogastric infection respectively. In contrast, Perforin-2-sufficient littermates clear the infection. Perforin-2 is a transmembrane protein of cytosolic vesicles -derived from multiple organelles- that translocate to and fuse with bacterium containing vesicles. Subsequently, Perforin-2 polymerizes and forms large clusters of 100 Å pores in the bacterial surface with Perforin-2 cleavage products present in bacteria. Perforin-2 is also required for the bactericidal activity of reactive oxygen and nitrogen species and hydrolytic enzymes. Perforin-2 constitutes a novel and apparently essential bactericidal effector molecule of the innate immune system.

*For correspondence:
epodack@miami.edu

Competing interests:
See page 25

## Introduction

Multicellular eukaryotes deploy pore-forming proteins to disrupt the cellular integrity of bacterial pathogens and virally infected cells. The first immunologically relevant discovery of a pore-former was the spontaneous polymerization and refolding of the hydrophilic complement component C9 into a membrane-associated cylindrical complex (*Podack and Tschopp, 1982*; *Tschopp et al., 1982*). This finding resolved the question of the molecular nature of the membrane attack complex of complement (MAC) (*Humphrey and Dourmashkin, 1969*; *Mayer, 1972*; *Muller-Eberhard, 1975*; *Bhakdi and Tranum-Jensen, 1978*) where C5b-8 complexes, first assembled around membrane-bound C3b, trigger C9 to polymerize and form 100 Å pores in bacterial surfaces (*Schreiber et al., 1979*; *Podack and Tschopp, 1982*; *Tschopp et al., 1982*).

The recognition that a single protein species, C9, was able to form pores by polymerization suggested the possibility that cytotoxic lymphocytes may be equipped with a similar pore-forming protein. Analysis of natural killer (NK) cells and cytotoxic T lymphocytes (CTL) identified Perforin-1 as

**eLife digest** An effective defense against foreign invaders is fundamental to an organism's survival. It is likely that immunity began to develop shortly after the emergence of Earth's first single-celled organisms and a remnant of that distant past still exists in our present day immune system in the form of Perforin-2. This ancient protein has been highly conserved throughout evolution from sea sponges to humans. Some studies have suggested that Perforin-2 may have an antimicrobial role in invertebrates (including clams, mussels, and snails) and fish. However, its mechanism of killing and its role in the mammalian immune systems has remained largely unknown.

McCormack et al. now report that Perforin-2 is a crucial component of host defense against a wide spectrum of infectious bacteria in both mice and humans. This was shown when mice lacking Perforin-2 died from bacterial infections that are not normally lethal. Somewhat unexpectedly, other bactericidal molecules were also found to be less effective in the absence of Perforin-2. This indicates that Perforin-2 is required for the activity of multiple aspects of the mammalian immune system.

McCormack et al. demonstrated that Perforin-2 kills by punching holes in bacteria. Unlike other pore-forming proteins that are only present in specific cells, all mammalian cells can express Perforin-2. McCormack et al. also showed that when Perforin-2 is produced at optimal levels, cells are able to combat otherwise lethal, drug-resistant bacteria, including methicillin resistant *Staphylococcus aureus* (MRSA). This means that Perforin-2 provides a rapid self-defense mechanism for cells against bacterial invaders. The protein's dual role as a pore-forming protein and a supporter of other antibacterial molecules is unprecedented. In the future, these findings could inform the development of treatments that activate and optimize Perforin-2 production to target and eradicate bacterial infections.

the pore-forming killer protein for virus-infected cells and tumor cells (*Dennert and Podack, 1983*; *Podack and Dennert, 1983*; *Blumenthal et al., 1984*). Sequence alignment of Perforin-1 and C9 identified a conserved domain, named the Membrane Attack Complex/Perforin (MACPF) domain in reference to its founding members (*Lichtenheld et al., 1988*).

During polymerization, the MACPF-domains of individual protomers refold and expose an amphipathic helix that inserts into the targeted membranes (*Rosado et al., 2007*; *Baran et al., 2009*; *Kondos et al., 2010*; *Law et al., 2010*). The hydrophilic surface of the membrane-inserted portion of polymerizing MACPF forms the inner, hydrophilic lining of the nascent pore driving the displacement of hydrophobic membrane components. MACPF generated pores disrupt the innate barrier function of membranes and provide access for chemical or enzymatic effectors that finalize destruction of the target (*Schreiber et al., 1979*; *Masson and Tschopp, 1987*; *Trapani et al., 1988*; *Shiver et al., 1992*; *Smyth et al., 1994*).

Macrophage Expressed Gene 1 (MPEG1) is the most recently identified protein with a MACPF-domain (*Spilsbury et al., 1995*). We renamed the new MACPF-containing protein Perforin-2 when we confirmed that it also was a pore forming protein. Evolutionary studies of Perforin-2, have demonstrated that Perforin-2 is one of the oldest eukaryotic MACPF members, present in early metazoan phyla including *Porifera* (sponges) (*D'Angelo et al., 2012*; *Wiens et al., 2005*; *McCormack et al., 2013a*; *McCormack et al., 2013b*). Orthologues of Perforin-2 are highly conserved throughout the animal kingdom (*Mah et al., 2004*; *Wiens et al., 2005*; *Wang et al., 2008*; *He et al., 2011*; *Kemp and Coyne, 2011*; *Green et al., 2014*). Recent studies in vertebrates (mammalia) demonstrate that expression of Perforin-2 is not limited to macrophages, as it was also detected in murine embryonic fibroblasts (MEF) and human epithelial cells after bacterial infection (*Fields et al., 2013*; *McCormack et al., 2013a*) suggesting that Perforin-2 expression is tied to antibacterial activity. Similarly, in Zebrafish one of its two isoforms, MPEG1.2, is induced following bacterial infection and limits bacterial burden (*Benard et al., 2015*).

Here we show that Perforin-2 is a major antibacterial effector protein of the innate immune system in phagocytic and in tissue forming cells. Perforin-2 is an essential innate effector protein that kills gram-positive, gram-negative, and acid-fast bacteria. The absence of Perforin-2 enables survival of pathogenic bacteria in vitro and systemic dissemination in vivo indicating that expression of Perforin-2 in professional phagocytes and in parenchymal cells is required to eliminate pathogenic bacteria in

vitro and in vivo. We demonstrate that Perforin-2 can polymerize to form pores visible by negative staining transmission electron microscopy in bacterial surfaces. The presence of Perforin-2 potentiates the antibacterial activity of other known effectors including reactive oxygen and nitrogen species. In our accompanying manuscript we report some of the molecular mechanisms of Perforin-2 activation and describe how a bacterial virulence factor blocks Perforin-2 function.

## Results

### Perforin-2-deficient neutrophils and macrophages are unable to kill pathogenic bacteria, including *Mycobacterium tuberculosis*

Professional phagocytes avidly ingest and kill bacteria. To elucidate the contribution of Perforin-2 towards their bactericidal activity, we compared professional phagocytes from Perforin-2 deficient mice with Perforin-2 heterozygous and wild-type phagocytes. Perforin-2-deficient murine peritoneal exudate macrophages (PEM), neutrophils, and bone marrow-derived macrophages (BMDM) are unable to kill three different species of Mycobacteria (*Mycobacterium smegmatis, Mycobacterium avium, M. tuberculosis*), as indicated by significant intracellular bacterial replication in *MPEG1* (Perforin-2) −/− compared to +/+ or +/− phagocytes (*Figure 1A–C*, *Figure 1—figure supplement 1*). Although BMDM express Perforin-2 constitutively, they must be activated with IFN and LPS in order to mediate Perforin-2-dependent growth inhibition of *M. tuberculosis* (Mtb) (*Figure 1—figure supplement 2*). This suggest that the destruction of Mtb requires both the expression and activation of Perforin-2.

We also used Perforin-2 siRNA to ablate Perforin-2 in cells in vitro. To exclude the possibility of off-target protein effects by Perforin-2-siRNA, we performed complementation assays in BV2-microglia with C-terminally tagged Perforin-2-RFP (*Figure 1D–G*). Endogenous Perforin-2 was silenced (*Figure 1H*, lane 2) with siRNA specific for the 3′-untranslated sequence and the cells were complemented by transfection with siRNA-resistant Perforin-2-RFP (*Figure 1H*, lane 1). Only Perforin-2-RFP but not control RFP transfection restored bactericidal activity. The data indicated that RFP-tagged Perforin-2 was fully active and that siRNA ablation of Perforin-2 has negligible off-target effects on bactericidal activity. The results suggested that Perforin-2 is critical for intracellular bacterial killing.

We also sought to determine the role of Perforin-2 in human professional phagocytes. Human macrophages and neutrophils express Perforin-2 protein constitutively (*Figure 1—figure supplement 3*). Human monocyte derived macrophages (MDM) efficiently killed intracellular MRSA. Perforin-2 knockdown by siRNA abrogated killing of MRSA and resulted in MRSA replication (*Figure 1 I*). In addition, we used the human promyelocytic cell line HL-60 that differentiates into Perforin-2-expressing neutrophils upon treatment with retinoic acid (RA). Perforin-2 siRNA silencing in RA-differentiated HL-60 cells abolished Perforin-2 protein expression (*Figure 1—figure supplement 3*) and was associated with intracellular replication of *Salmonella typhimurium*, MRSA, and *M. smegmatis* (*Figure 1—figure supplement 4*). In contrast, scramble-transfected controls continued to express endogenous Perforin-2 and the number of recovered bacteria was reduced over several hours. This result suggested that Perforin-2 was also required for the killing of bacteria in human neutrophils (*Figure 1—figure supplement 4*).

In summary, the results indicate that professional murine and human phagocytes require Perforin-2 to kill phagocytosed bacteria. This finding raised the question of the function of the other known bactericidal mediators in relation to Perforin-2.

### Reactive oxygen and nitrogen species enhance the bactericidal activity of Perforin-2 but have little microbicidal activity when Perforin-2 is absent

Reactive oxygen (ROS) and reactive nitrogen (RNS) species are recognized for their bactericidal activity in phagocytic cells. PEM activated by IFN-γ and LPS produce both families of effectors (*Figure 2—figure supplement 1*). We used the well-characterized chemical inhibitors N-acetyl-L-cysteine (NAC) and L-N^G-nitroarginine methyl ester (L-NAME) to block ROS and nitric oxide (NO) respectively as Perforin-2 and ROS or Perforin-2 and RNS knockout animals are not currently available (*Vazquez-Torres et al., 2000*; *Mantena et al., 2008*; *Sohn et al., 2011*).

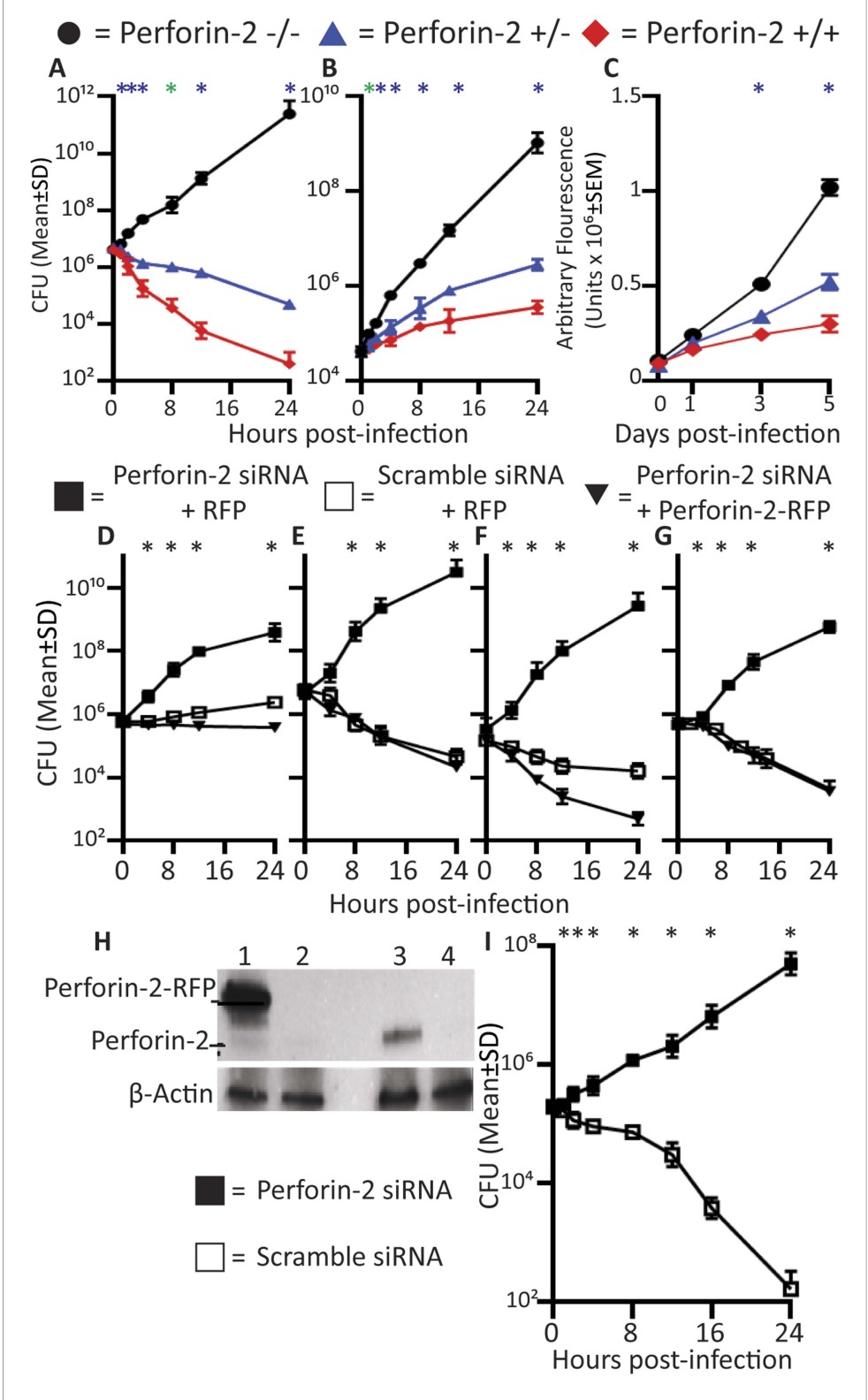

Figure 1. Perforin-2 deficiency or siRNA knockdown abrogates intracellular killing of pathogenic bacteria. (A-C) Perforin-2 knockout, heterozygous, and wild-type macrophages and neutrophils were infected with Mycobacterium species (A) PEM infected with *Mycobacterium smegmatis*, (B) Neutrophils infected with *Mycobacterium avium*, and (C) BMDM infected with *Mycobacterium tuberculosis*. (D-H) Perforin-2 knockdown can be complemented in BV2

*Figure 1. continued on next page*

*Figure 1. Continued*

microglia cells infected with (**D**) *M. avium*, (**E**) *M. smegmatis*, (**F**) *Salmonella typhimurium*, and (**G**) MRSA. (**H**) Western blot demonstrating protein levels after complementation: BV2 transfected with (Lane 1) Perforin-2-RFP and Perforin-2 siRNA, (Lane 2) RFP and Perforin-2 siRNA, (Lane 3) RFP and Perforin-2 scramble siRNA, and (Lane 4) Perforin-2 siRNA alone. In western blots, Perforin-2-RFP is detected as a 105 kD band compared to the 72 kD band seen for endogenous Perforin-2 (lane 1 and 3 respectively). (**I**) Human MDM infection with MRSA. ◆= *MPEG1* (Perforin-2) wild-type cells (+/+), ▲= *MPEG1* (Perforin-2) heterozygous cells (+/−), ● *MPEG1* (Perforin-2) knockout cells (−/−). ■= RFP + Perforin-2 siRNA transfected cells, □= RFP + scramble siRNA transfected cells. ▼= Perforin-2-RFP + Perforin-2 siRNA transfected cells. One-way ANOVA with Tukey's multiple comparisons post-hoc test was used for A–G. (**A–C**) *$p < 0.05$ between Perforin-2 knockout:Perforin-2 wild-type cells; *$p < 0.05$ between Perforin-2 knockout:Perforin-2 wild-type and Perforin-2 knockout:Perforin-2 heterozygous cells. (**D–G**) *$p < 0.05$ between RFP + Perforin-2 siRNA: RFP + scramble siRNA and RFP + Perforin-2 siRNA:Perforin-2-RFP + Perforin-2 siRNA. (**I**) *$p < 0.05$ multiple t-tests with post-hoc correction for multiple comparisons using the Holm-Sidak method.

The following figure supplements are available for figure 1:

**Figure supplement 1**. Perforin-2 genotype reflects total amount of Perforin-2 protein produced.

**Figure supplement 2**. Perforin-2 dependent growth inhibition of Mtb in BMDM requires activation by LPS and IFN-γ.

**Figure supplement 3**. Perforin-2 is present in human MDM and PMN and can be knocked down in RA treated HL60.

**Figure supplement 4**. Human HL-60-differentiated neutrophils require Perforin-2 to eliminate pathogens.

First, we established that the addition of the inhibitors reduced levels of ROS and NO produced by activated PEMs (*Figure 2—figure supplement 1*). The role of endogenous ROS and NO on cellular bactericidal activity in PEM in the presence and absence endogenous of Perforin-2 was assessed in two complementary ways. First, we assessed the effect of chemical inhibitors of ROS and NO on killing of intracellular wild-type *S. typhimurium* (*Figure 2A–D*). ROS is known to be active and produced during the first 4 hr after *S. typhimurium* infection in PEM (*Mastroeni et al., 2000*). In Perforin-2 deficient PEM, *S. typhimurium* replicated equally well regardless of ROS inhibition (*Figure 2B*). This suggests ROS had minimal influence on intracellular replication of *S. typhimurium* in the absence of Perforin-2. In contrast, with PEM that express Perforin-2, the inhibition of endogenous ROS by NAC enables *S. typhimurium* to replicate significantly more than mock treatment during the first 4 hr after infection, suggesting that ROS in combination with Perforin-2 helps to restrain *S. typhimurium* replication during this period (*Figure 2A*).

Inhibition of endogenous NO production with L-NAME in the presence of Perforin-2 also allowed increased intracellular *S. typhimurium* replication beginning several hours post-infection (*Figure 2C*) coinciding with the known time period required for onset of NO production (see *Figure 2—figure supplement 1*) (*Mastroeni et al., 2000*). As with ROS, endogenous NO had little effect on *S. typhimurium* replication in the absence of Perforin-2 (*Figure 2D*). The results indicate that although ROS and NO contribute towards the intracellular killing of bacterial pathogens, their bactericidal activity is largely dependent upon the presence of Perforin-2.

To validate that the cooperation between Perforin-2 and ROS and NO was not specific to *S. typhimurium*, we repeated the above with *M. smegmatis*. The cooperative activity of Perforin-2 with ROS and NO was also evident in killing of *M. smegmatis* in PEM (*Figure 2—figure supplement 2*). Pharmacologic inhibitors significantly increased bacterial survival only when Perforin-2 was present, with little additional effect when Perforin-2 was absent. This data suggested that the critical importance of Perforin-2 in facilitating ROS and NO activity is not specific to only *S. typhimurium*.

To further investigate the dependence of ROS and NO bactericidal activity upon Perforin-2 we utilized *S. typhimurium* mutants lacking the periplasmic superoxide dismutase (*sodC1*) or flavohemoglobin (*hmpA*) genes, which neutralize ROS or NO, respectively, in intracellular killing assays. (*Stevanin et al., 2002*; *Uzzau et al., 2002*; *Krishnakumar et al., 2004*; *Prior et al., 2009*). If ROS and NO activity is dependent upon Perforin-2 we hypothesized that bacterial ROS and NO

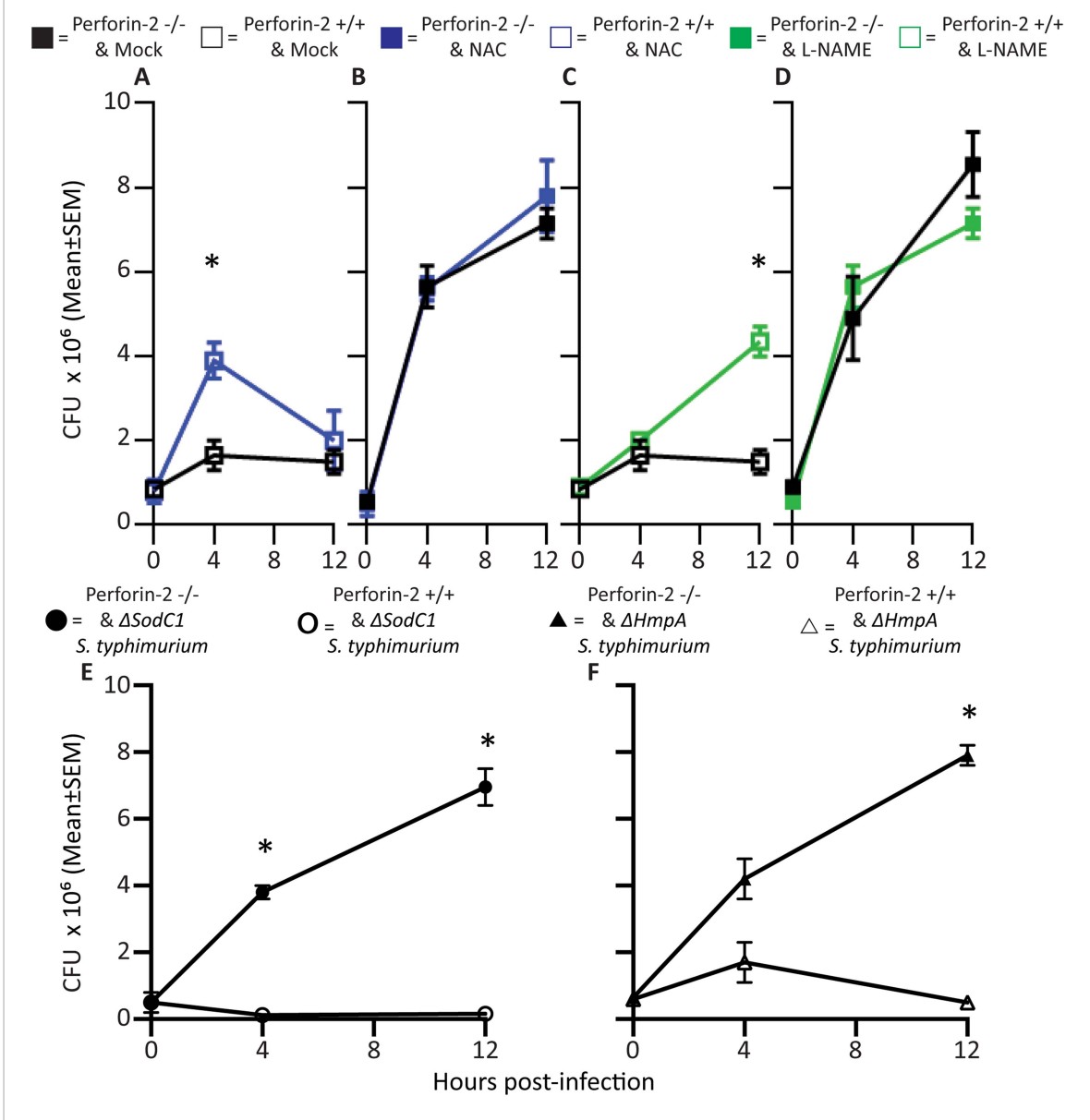

**Figure 2**. Antimicrobial compounds (ROS and NO) enhance Perforin-2 mediated killing of S. typhimurium by PEM but have limited activity in the absence of Perforin-2. (A–D) Wild-type *S. typhimurium* infection of PEMs isolated from either *MPEG1* (Perforin-2) +/+ (**A**, **C**), or *MPEG1* (Perforin-2) −/− mice (**B**, **D**). Non-filled symbols indicated *MPEG1* (Perforin-2) +/+ PEMs; whereas filled symbols are *MPEG1* (Perforin-2) −/− PEMs. Cells were incubated with NAC (blue line), NAME (green line), or mock (black line). To assess bacterial resistance mechanisms against these effectors, (**E**) SodC1 or (**F**) HmpA knockout *S. typhimurium* were used to infect *MPEG1* (Perforin-2) −/− or +/+ PEMs. The above experiments were conducted with six biologic replicates and are representative of four independent experiments. Statistical analysis was performed utilizing multiple T-tests with correction for multiple comparisons using the Holm-Sidak method. *$p < 0.05$.

The following figure supplements are available for figure 2:

**Figure supplement 1**. Nitrite and reactive oxygen production in PEMs following addition of inhibitors.

**Figure supplement 2**. Antimicrobial compounds (ROS and NO) enhance Perforin-2 mediated killing of *M. smegmatis* by PEM.

defense mechanisms would be unnecessary in Perforin-2 knockout cells. Unlike other superoxide dismutases that protect bacteria from oxygen radicals produced intracellularly as a byproduct of cellular respiration, SodC1 is a periplasmic superoxide dismutase. In vivo, *sodC1* mutants are

significantly attenuated relative to wild-type *S. typhimurium* (*De Groote et al., 1997*; *Fang et al., 1999*; *Krishnakumar et al., 2004*). Flavohemoglobin acts by either catalyzing an $O_2$-dependent denitrosylase reaction converting NO to a nitrate ion or $N_2O$, or an anoxic reductive reaction forming $NO^-$. As with SodC1, in vivo and in vitro studies substantiate the role of HmpA with significant attenuation observed with HmpA deficient bacteria (*Stevanin et al., 2002*; *Bang et al., 2006*). In the presence of Perforin-2, SodC1-deficient *S. typhimurium* were killed more efficiently than wild-type *S. typhimurium*. However, SodC1-deficient *S. typhimurium* replicate similar to wild-type *S. typhimurium* when Perforin-2 was absent (*Figure 2E*). Similarly, *hmpA* mutants are more susceptible to killing by NO than wild-type bacteria, but only when Perforin-2 is present. In the absence of Perforin-2, Flavohemoglobin does not enhance the survival and replication of wild-type *S. typhimurium* relative to the *hmpA* mutant.

Thus, both chemical and genetic analyses indicate that Perforin-2 is required for the bactericidal activity of ROS and NO in macrophages.

## Perforin-2 is required for bactericidal activity of parenchymal, tissue forming cells

The induction of Perforin-2 in certain parenchymal cells was reported previously (*Fields et al., 2013*; *McCormack et al., 2013*). We expanded this analysis for many human and murine primary cells and established cell lines, ranging from epithelial to endothelial cells, from astrocytes to myoblasts, and from neural cells to secretory cells (*Table 1*, *Table 2*). Every cell type derived from ectodermal, neuroectodermal, endodermal, or mesodermal lineage tested to date is able to express Perforin-2 message either constitutively or after type I or II IFN induction. *Table 1* (murine cells) and *Table 2* (human cells) summarize these results while their respective supplements (*Supplementary files 1, 2*) show the inducibility of Perforin-2's mRNA and protein (qPCR of ΔCT of Perforin-2 normalized to GAPDH and western blot analysis). Moreover, all cell types analyzed (54 out of 54) are able to kill bacteria in an in vitro bactericidal assay when Perforin-2 is expressed. When infection occurs prior to Perforin-2 induction or when Perforin-2 is siRNA-ablated or genetically deficient using the above assay, bacteria were not killed by cells and consequently replicate. In contrast, cells that express Perforin-2 were bactericidal. These results suggest that Perforin-2 can be expressed ubiquitously to defend cells against bacterial invasion.

Perforin-2 siRNA knockdown was used to determine its contribution towards intracellular killing of bacteria by IFN-induced murine and human parenchymal cells. Although IFN induces hundreds of antimicrobial genes in addition to Perforin-2, silencing of Perforin-2 alone was sufficient to cause bacterial replication. Without exception, Perforin-2 expression and function were essential for killing a diverse array of intracellular pathogenic bacteria by parenchymal or phagocytic cells. Examples of bactericidal activity include human vascular endothelial cells (HUVEC); human pancreatic cells (MIA-PaCa-2); human uroepithelial cells (UM-UC-9); murine ovarian epithelial cells (MOVCAR 5009); murine colon epithelial cells (CT26); and murine cardiac myoblasts (C2C12), respectively (*Figure 3* and *Supplementary file 3*). Examples of human and murine siRNA-mediated Perforin-2 protein knockdown include HUVECs and myoblasts (*Figure 3*, *Supplementary file 4*). To certify that Perforin-2 siRNA targeting was specific, siRNA resistant Perforin-2-RFP was utilized to complement Perforin-2 siRNA in parenchymal tissue forming cells. *Figure 3—figure supplement 1* highlights representative examples of Perforin-2 complementation in myoblasts, intestinal epithelial cells, and PEM.

To further validate the requirement of Perforin-2 for bactericidal activity in non-phagocytic cells, we used genetically deficient MEFs obtained from *MPEG1* (Perforin-2) +/+, +/−, and −/− littermates. After overnight induction with IFN-γ, *MPEG1* (Perforin-2) +/+ MEFs eliminate MRSA. In contrast, IFN-γ treated Perforin-2 −/− MEFs enable MRSA to replicate. Heterozygous MEFs had intermediate bactericidal activity (*Figure 3D*) suggesting a gene dose effect of Perforin-2.

Keratinocytes (Kc) represent the first cellular barrier to infection in the skin (*Song et al., 2002*; *Mempel et al., 2003*; *Bernard and Gallo, 2011*). Unlike other parenchymal cells that need to be induced, primary human Kc express Perforin-2 constitutively. The identity of human Perforin-2 in Kc was confirmed by western blotting (*Figure 3—figure supplement 2*). Densitometry analysis suggests similar Perforin-2 protein levels in MDM and Kc (*Figure 3—figure supplement 2*). Expression could also be silenced with human Perforin-2 siRNA (*Figure 3—figure supplement 2*, lane 2), but not with

**Table 1**. Murine perforin-2 expression

| Cell type: | Perforin-2 expression: |
| --- | --- |
| *Peritoneal macrophage* | Constitutive |
| *Bone marrow derived macrophage (BMDM)* | Constitutive |
| *Bone marrow derived dendritic cell (BMDC)* | Constitutive |
| BV-2 microglia cell line | Constitutive |
| Raw264.7 macrophage cell line | Constitutive |
| J774A.1 macrophage cell line | Constitutive |
| *Microglia* | Constitutive |
| *Neutrophil (peritoneum stimulation)* | Constitutive |
| *Neutrophil (bone marrow)* | Constitutive |
| *Gamma delta (γδ) T cell (from Skin)* | Constitutive |
| *Gamma delta (γδ) T cell (from Gut)* | Constitutive |
| *Gamma delta (γδ) T cell (from Vagina)* | Constitutive |
| *Marginal zone B cell* | Constitutive |
| *Keratinocyte (Back)* | Constitutive |
| *Intestinal epithelial cells* | Constitutive |
| *Splenocytes* | Constitutive |
| *OT1 CD8 T cell induced with TGFβ, RA, and IL2* | Constitutive |
| *OT1 CD8 T cell* | Inducible |
| *CD4 T cell* | Inducible |
| *B cell* | Inducible |
| *Astrocyte* | Inducible |
| *Neuron* | Inducible |
| Cath.a neuroblastoma cell line | Inducible |
| Neuro-2A neuroblastoma cell line | Inducible |
| *Adult CNS fibroblast* | Inducible |
| *Embryonic fibroblast* | Inducible |
| NIH 3T3 fibroblast cell line | Inducible |
| Balb/c 3T3 fibroblast cell line | Inducible |
| C2C12 myoblast cell line | Inducible |
| *Neonatal ventricular myocytes* | Inducible |
| CMT-93 rectal carcinoma cell line | Inducible |
| CT26 colon carcinoma cell line | Inducible |
| B16-F10 melanoma cell line | Inducible |
| B16-F0 melanoma cell line | Inducible |
| MOVCAR 5009 ovarian cancer cell line | Inducible |
| MOVCAR 5447 ovarian cancer cell line | Inducible |
| LL/2 Lewis lung carcinoma cell line | Inducible |
| ED-1 lung adenocarcinoma cell line | Inducible |

Italics: *Ex vivo* primary cells utilized for analysis.

scramble siRNA (**Figure 3—figure supplement 2**, lane 3). The inhibition of Perforin-2 expression abrogated the bactericidal activity of Kc (**Figure 3E**) that were able to kill MRSA, irrespective of prior IFN activation (**Figure 3F**).

**Table 2**. Human peforin-2 expression

| Cell type: | Perforin-2 expression: |
|---|---|
| *Monocyte derived macrophage (MDM)* | Constitutive |
| *Monocyte derived dendritic cell (MDC)* | Constitutive |
| *PBMC isolated NK cell* | Constitutive |
| *Polymorphonuclear granulocyte (neutrophil)* | Constitutive |
| HL-60 promyelocyte cell line RA differentiated to PMN | Constitutive |
| HL-60 cell line PMA differentiated to Macrophage | Constitutive |
| *Fetal keratinocyte* | Constitutive |
| *Adult keratinocyte* | Constitutive |
| PMA differentiated Thp-1 monocyte cell line | Constitutive |
| NK-92 cell line | Constitutive |
| *Normal colon biopsy* | Constitutive |
| *Normal skin biopsy* | Constitutive |
| *Umbilical endothelial cell (HUVEC)* | Inducible |
| HeLa cervical carcinoma cell line | Inducible |
| A2EN endocervical epithelial cell line | Inducible |
| UM-UC-3 bladder cancer cell line | Inducible |
| UM-UC-9 bladder cancer cell line | Inducible |
| CaCo-2 colorectal carcinoma cell line | Inducible |
| HEK293 embryonal kidney cell line | Inducible |
| MIA-PaCa-2 pancreatic cancer cell line | Inducible |
| *Skin fibroblast* | Inducible |
| Thp-1 monocyte cell line | Inducible |
| HL-60 promyelocyte cell line | Inducible |
| OVCAR3 ovarian carcinoma cell line | Inducible |
| A549 alveolar adenocarcinoma cell line | Inducible |
| U-1752 bronchiolar epithelial cell line | Inducible |
| Jeg-3 placental choriocarcinoma cell line | Inducible |

Italics: *Ex vivo* primary cells utilized for analysis.

Cumulatively, our results suggest that Perforin-2 is an effector for cellular defense against pathogenic bacteria in professional phagocytes and in other cells. The ubiquity of Perforin-2 expression suggests a critical importance in cellular defenses of many, if not all, tissue against pathogenic bacteria. The findings raise the question of the molecular mechanisms by which Perforin-2 exerts its potent bactericidal function.

## Perforin-2 accumulates in membranes enclosing bacteria and is associated with bacterial lysis

The MACPF domain of Perforin-2 suggests that it is a pore-forming protein similar to the pore-formers of complement (C9) and cytotoxic lymphocytes (Perforin-1) (*Podack and Tschopp, 1982*; *Dennert and Podack, 1983*; *Law et al., 2010*). In analogy to C9 and perforin-1, pore-formation by the MACPF domain of Perforin-2 on the bacterial surface may constitute the lethal hit. However, Perforin-2, unlike C9 and Perforin-1, is anchored in membrane vesicles with its MACPF domain predicted to reside inside the vesicle or outside on the plasma membrane (*Figure 4A*). Therefore we decided to study the cell biology of Perforin-2 in resting cells and following bacterial infection.

In lieu of antibodies capable of detecting native endogenous Perforin-2, as these reagents are not currently available, we transfected c-terminal tagged Perforin-2–RFP or–GFP into cells in which

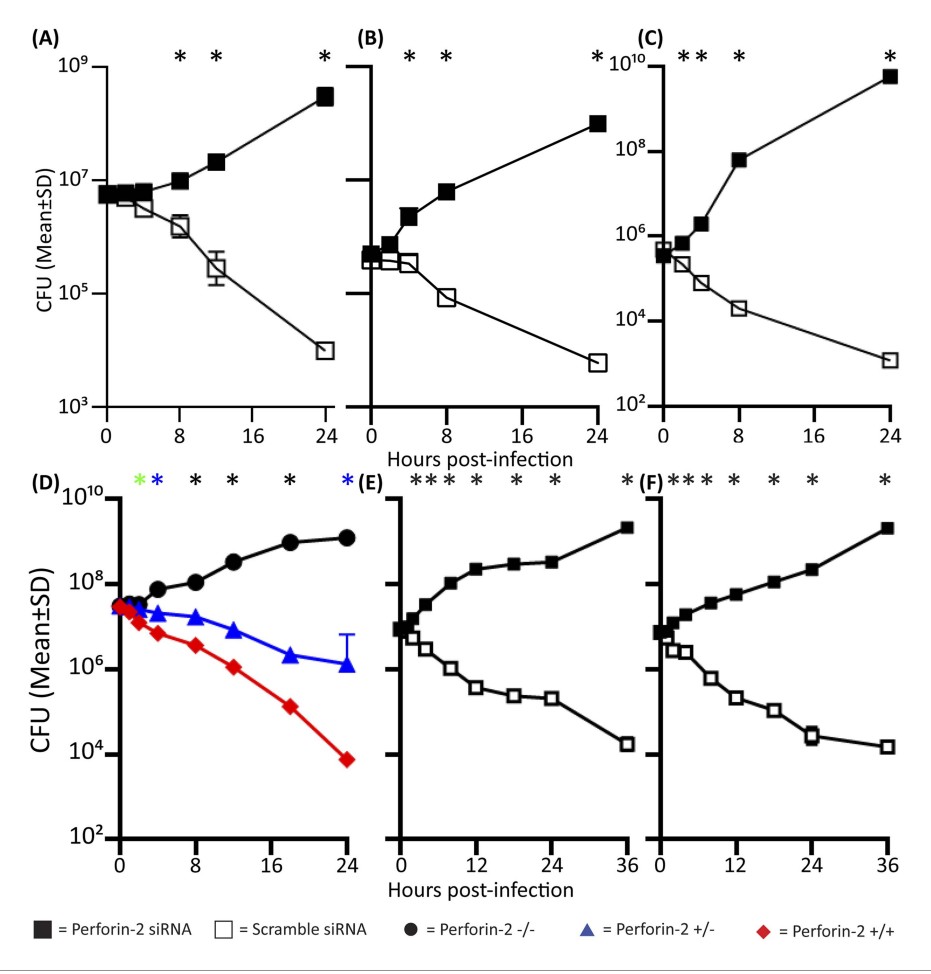

**Figure 3**. Perforin-2 significantly contributes to intracellular killing in non-hematopoietically derived cells. One day prior to the infection, cells were transfected with either a pool of scramble (□) or Perforin-2 specific (■) siRNA and 14 hr prior to the infection induced with IFN-γ. (**A**) HUVEC cells infected with *M. smegmatis*, (**B**) MIA-PaCa-2 cells infected with *S. typhimurium*, (**C**) UM-UC-9 infected with MRSA, (**D**) Perforin-2 MEF infected with MRSA, (**E**) Human Kc infected with MRSA induced with IFN-γ, (**F**) Human Kc infected with MRSA with no IFN-γ induction. ◆= *MPEG1* (Perforin-2) +/+, ▲= *MPEG1* (Perforin-2) +/−, ●= *MPEG1* (Perforin-2) −/−. (**A–C**, **E**, **F**) The above graphs contain 5–9 biologic replicates, and are representative of 3–7 independent experiments. Statistical analysis was performed utilizing multiple T-tests with correction for multiple comparisons using the Holm-Sidak method. *p < 0.05. (**D**) One-way ANOVA with Tukey post-hoc multiple comparisons. *p < 0.05 between Perforin-2 knockout:Perforin-2 wild-type mice *p < 0.05 between Perforin-2 knockout:Perforin-2 wild-type and Perforin-2 knockout:Perforin-2 heterozygous mice. *p < 0.05 between Perforin-2 knockout: Perforin-2 wild-type, Perforin-2 knockout:Perforin-2 heterozygous, and Perforin-2 heterozygous:Perforin-2 wild-type.

The following figure supplements are available for figure 3:

**Figure supplement 1**. Perforin-2 knockdown is complementable and is able to replicate endogenous Perforin-2 bactericidal function.

**Figure supplement 2**. Perforin-2 protein expression in human primary keratinocytes after knockdown compared to human monocyte derived macrophages (MDM).

endogenous Perforin-2 was knocked down. The Perforin-2 fusion proteins were functionally active as shown above in complementation studies (*Figure 1*, *Figure 3—figure supplement 1*). In a murine macrophage cell line (RAW264.7) tagged murine Perforin-2 was localized to ER, Golgi, and early endosomes based on its overlay with ER tracker, GM130, and EEA1 respectively (*Figure 4*, *Figure 4—figure supplement 1*).

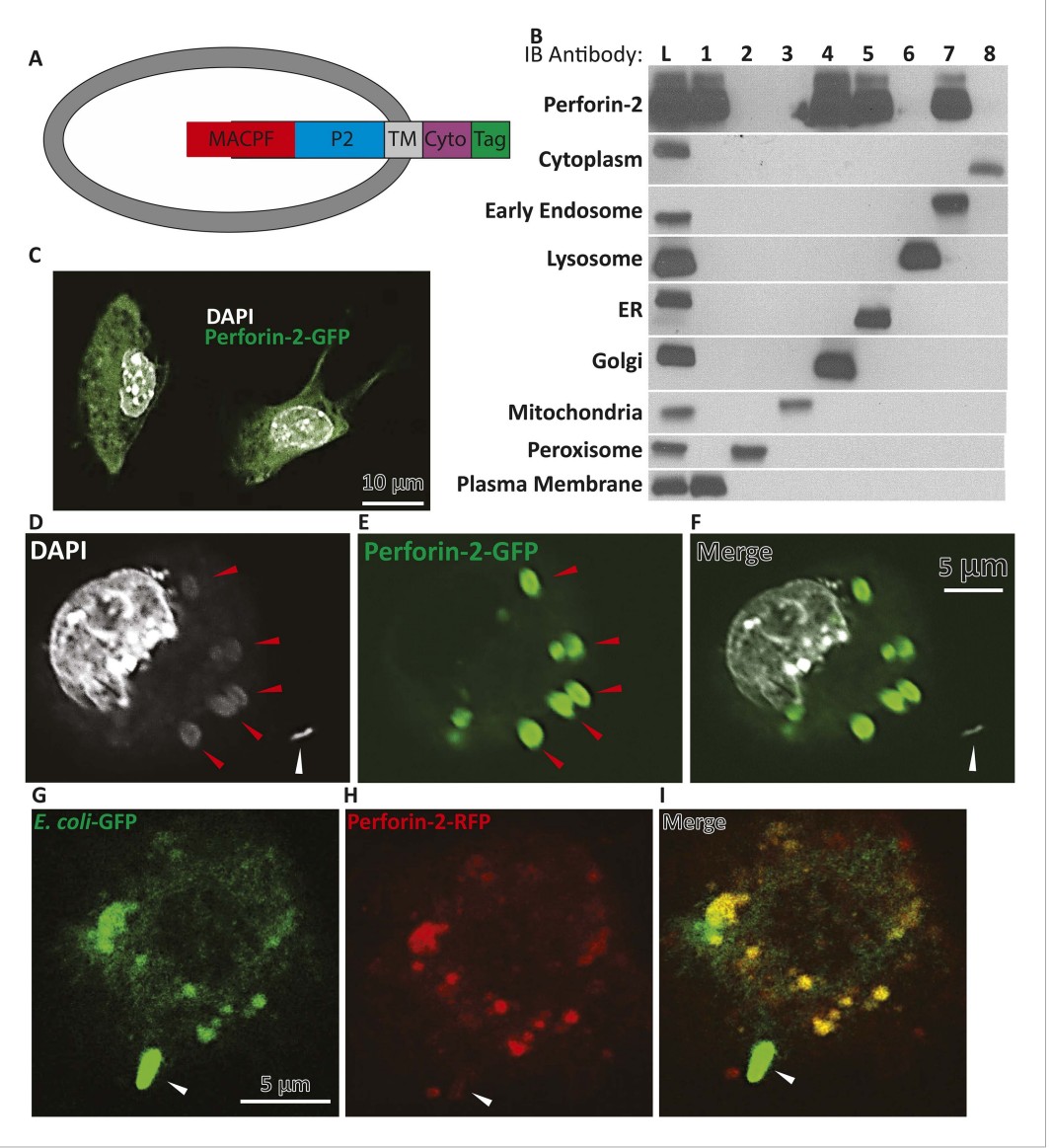

Figure 4. Endogenous Perforin-2 is located in intracellular sites allowing for rapid translocation to bacteria. (A) Schematic demonstrating proposed orientation of Perforin-2 in vesicles. (B) Fractionation results of endogenous Perforin-2 from human macrophages. (Lane L) is a post-nuclear lysate control, (Lane 1–8) are individual fractions corresponding with specific indicated organelles. (C) Overexpression of murine Perforin-2-GFP in murine BV2 microglial cells. (D–F) Confocal images taken 5 min after *S. typhimurium* infection in Perforin-2-GFP + Perforin-2 siRNA transfected BV2 cells. White arrows denote extracellular *S. typhimurium*, red arrows highlight a DNA cloud corresponding with *S. typhimurium* (D) DAPI only, (E) Perforin-2-GFP only, (F) Merge of DAPI and Perforin-2-GFP. (G–I) Confocal images taken 5 min after *Escherichia coli*-GFP infection in Perforin-2-RFP + Perforin-2 siRNA transfected BV2 cells. Arrows point to extracellular *E. coli*-GFP that has made contact but is still extracellular with normal bacilli morphology maintained. (G) *E. coli*-GFP only, (H) Perforin-2-RFP only, and (I) merge *E. coli*-GFP and Perforin-2-RFP. Fractions in B were probed as follows: Cytoplasm—MEK1/2; Early Endosome—EEA1; Lysosome—Lamp1; ER—calreticulin; Golgi—Golgin-97; Mitochondria—Prohibitin; Peroxisome—Catalase; Plasma Membrane—Cadherin.

The following figure supplements are available for figure 4:

**Figure supplement 1**. Perforin-2-RFP colocalizes with ER, Golgi, and early endosomes.

*Figure 4. Continued*

**Figure supplement 2**. S. typhimurium infection of macrophages with Perforin-2 and DAPI localization through the cell (multiple Z-sections).

**Figure supplement 3**. Perforin-2-RFP colocalizes with *E. coli*-GFP within minutes of infection.

To validate the results obtained thus far with the Perforin-2 fusion proteins, we also analyzed endogenous human Perforin-2 in fractions of post-nuclear homogenates of human monocytic leukemia (THP-1) differentiated to a macrophage phenotype (*Figure 4B*). Upon density gradient ultracentrifugation, Perforin-2 co-sedimented with membrane vesicles staining with markers for ER, Golgi, early endosome, or plasma membrane. Perforin-2 was not detected in fractions containing mitochondrial, peroxisomal, or lysosomal membranes, nor in the cytoplasm. As Perforin-2 localizes to the same organelles following co-sedimentation of endogenous Perforin-2 or imaging after overexpressed c terminal tagged Perforin-2, it is unlikely that the tag will disrupt Perforin-2 localization following bacterial challenge.

In non-infected murine BV2 microglial cells, Perforin-2-GFP vesicles were distributed in a membrane pattern throughout the cell (*Figure 4B,C*). Following *S. typhimurium* infection, Perforin-2-GFP redistributed and accumulated, within minutes, on endosomal/phagosomal bodies (*Figure 4D–F*, red arrows) that contained diffuse DNA as indicated by DAPI staining of phagocytosed *S. typhimurium* as compared to an intact extracellular *S. typhimurium* bacillus with its typical rod-like morphology (white arrow). *Figure 4—figure supplement 2* demonstrated a series of z-sections through the same cell suggesting that phagocytosis of several *S. typhimurium* bacilli had occurred. These images suggested that bacterial DNA was released from *S. typhimurium* killed by Perforin-2-GFP carried in vesicular membranes.

To directly demonstrate the presence of bacteria in Perforin-2-RFP-containing endosomes, we infected microglial cells expressing Perforin-2-RFP with *Escherichia coli* expressing GFP. Fluorescent imaging indicated that Perforin-2-RFP is largely concentrated on phagosomes that contain *E. coli*-GFP (*Figure 4G–I*). Phagocytosis and killing of *E. coli*-GFP ranged from incipient ingestion of largely intact *E. coli*-GFP to phagosomes containing green fluorescence that may have been released from killed *E. coli*-GFP. Merging green and red fluorescence shows co-localization. Regions with only green or red fluorescence suggested that a green bacterium being phagocytosed had not yet fused with the Perforin-2-RFP expressing endosome. *Figure 4—figure supplement 3* shows lower magnification field views.

Overall, these studies demonstrated that Perforin-2 is able to rapidly translocate to endosomal membranes, or in some cells may already be localized in these membranes in order to trap intracellular bacteria (upon fusion with phagosomes). During this process GFP is released from GFP synthesizing bacteria suggesting bacterial demise. Similarly, the appearance of diffuse bacterial DNA inside the vesicle suggests that integrity of the bacteria has been compromised. Owing to Perforin-2's highly conserved MACPF domain, we next were interested in elucidating if Perforin-2 and its MACPF domain can form pores.

## Perforin-2 forms large clusters of 100 Å pores in bacterial cell walls

To address the hypothesis that Perforin-2 can form pores in membranes akin to C9 of complement and Perforin-1 we used an artificial system by overexpressing Perforin-2 in HEK-293 cells, followed by the isolation of post-nuclear membranes, limited proteolysis, and their analysis by negative staining electron microscopy (EM). The cytoplasmic domain of Perforin-2 has a predicted trypsin cleavage site close to the N-terminus of its transmembrane domain which may be functionally important. We observed clusters of typical membrane pores of ~100 Å inner diameter in these membranes (*Figure 5A,B*) that were morphologically similar to poly-C9 and poly-Perforin-1 pores on bacterial and cell membranes (*Schreiber et al., 1979*; *Podack and Tschopp, 1982*; *Podack and Dennert, 1983*). Pores were not observed on any membranes in the absence of trypsin treatment. These experiments suggest that Perforin-2, putatively via its MACPF domain, is able to form clusters of regular polymeric pores on membranes akin to C9 of complement and Perforin-1. Although our methodology facilitated

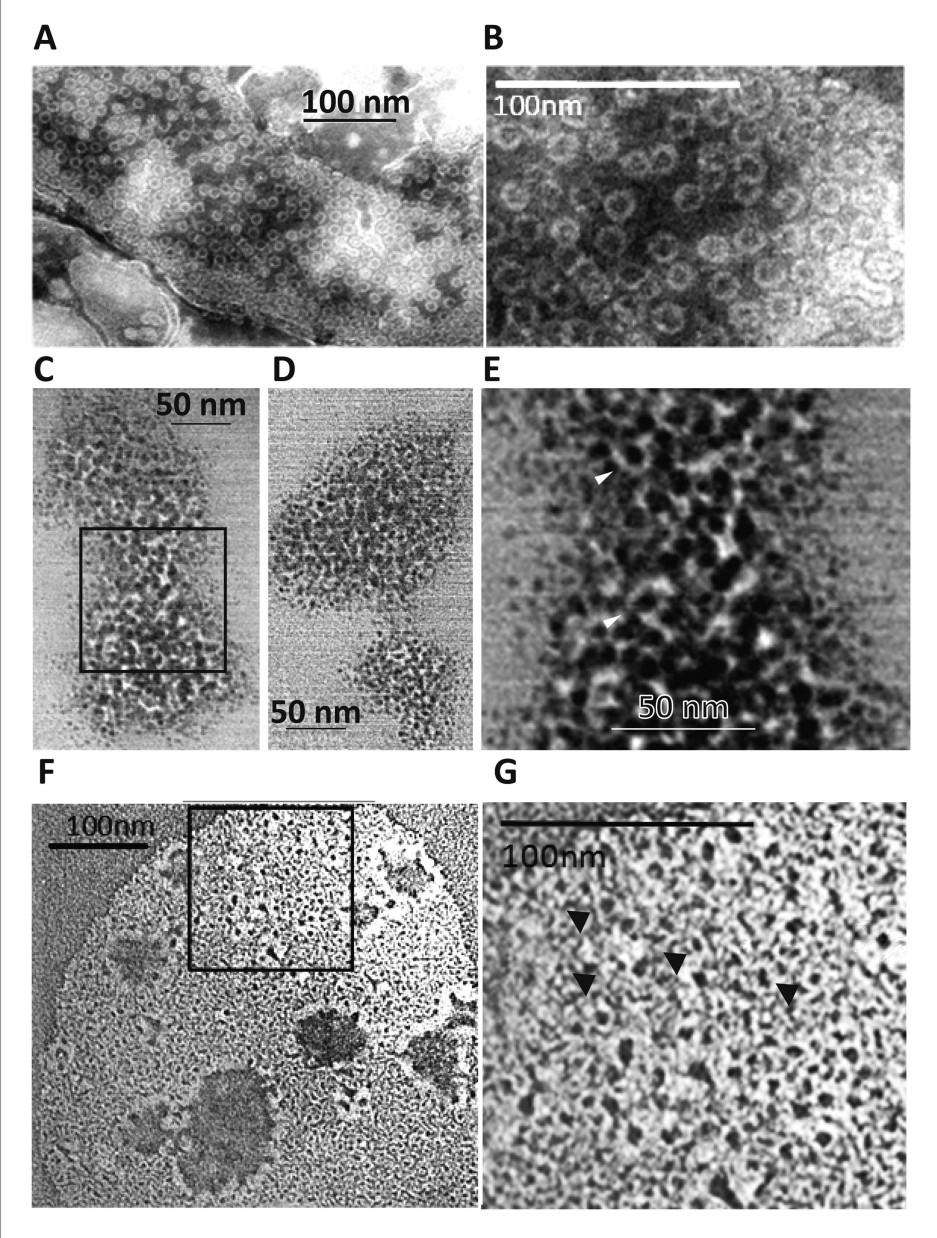

**Figure 5**. Perforin-2 forms pores in bacterial surfaces after infection and in Perforin-2 overexpressed eukaryotic membranes that are visible by negative stain transmission electron microscopy (TEM). (**A**, **B**) Electron micrograph of polymerized Perforin-2 membrane lesions from Perforin-2-GFP transfected HEK-293 cells, with Perforin-2 activated to form pores by trypsin digestion to the enriched membrane fraction. Panel A Demonstrates the quantity of pores on the Perforin-2 overexpressed membranes after trypsin activation. Panel B denotes a higher magnification to illustrate the uniform pore structure. (**C**-**G**) Perforin-2 wild-type MEFs were treated with IFN-γ for 14 hr, and infected with (**C**–**E**) MRSA or (**F**, **G**) *M. smegmatis*. After 5 hr the infected bacteria were isolated and imaged utilizing negative stain TEM. Arrows point to black, stain-filled pores on the bacterial cell wall surrounded by white, stain excluding borders created by polymerized Perforin-2. Round pores measure 8.5–10 nm inner diameter, the size typical for polymerized Perforin-2-pores. Panels **E** and **G** are close-up images of the boxed region in C and **F**. Blinded quantification of pore amount with different conditions is demonstrated in *Figure 6—figure supplement 1*.

The following figure supplement is available for figure 5:

**Figure supplement 1**. Quantification of pores from negative stain transmission electron microscopy.

visualization of poly Perforin-2 pores, we are not suggesting that Perforin-2 is typically activated by trypsin nor are we suggesting that eukaryotic membranes are typical targets of Perforin-2.

The MACPF domain has been shown to be the pore-forming domain of Perforin-1 and of the membrane attack complex of complement (*Hadders et al., 2007*; *Rosado et al., 2007*; *Slade et al., 2008*; *Law et al., 2010*). The physical proximity of the pore-forming MACPF domain of Perforin-2 to endocytosed bacteria suggested that bacteria killed inside the endosomes might also exhibit electron microscopic images typical for Perforin-2-lesions on bacterial surfaces/membranes. To test this hypothesis, we induced Perforin-2 expression in MEF overnight with IFN-γ and infected the following day with *M. smegmatis* or MRSA. Intracellular bacteria were then re-isolated from MEFs after 5 hr and bacterial surfaces were inspected by negative-staining electron microscopy (*Figure 5C–G*). Large patches of densely clustered pores of ca. 100 Å inner diameter are visualized on both MRSA and *M. smegmatis* surfaces with similar irregularities as those observed in complement lesions on *E. coli* surfaces (*Schreiber et al., 1979*). MRSA bacterial surfaces are more hydrophilic and show more uniform negative staining (*Figure 5C–E*) when compared to *M. smegmatis* samples that appear washed out with lower detail and contrast. This observed difference is due to the high hydrophobicity of the mycobacterial surface that repels negative stain except where it accumulates in the pore (*Figure 5F,G*) (*Noda and Kanemasa, 1986*; *Stokes et al., 2004*). Pores are only visible after bacteria were isolated from cells expressing Perforin-2. Pores were not present on control bacteria or after isolation of bacteria from Perforin-2 deficient cells as determined by blinded quantification of the pore-like structures (*Figure 5—figure supplement 1*). There were significantly greater numbers of pore-like structures (two or more orders of magnitude) associated with bacteria isolated from Perforin-2 expressing cells compared to Perforin-2 deficient cells. This indicates that there is a strong correlation between pore-formation and Perforin-2.

## Perforin-2 cleavage fragments are detectable in isolated bacteria after infection

Although it is not yet technically possible to prove definitively that the visualized pores are poly-Perforin-2, the existence of Perforin-2 pores is contingent upon a physical association between Perforin-2 and bacterial cells. Since we have a panel of antibodies that recognize denatured Perforin-2, we reasoned that it is possible to prove or disprove the latter. To accomplish this, we utilized five antibodies raised against peptides from different regions of human Perforin-2 (*Figure 6—figure supplement 1*). Perforin-2–/– MEFs were transfected with either GFP or human Perforin-2-GFP plasmids and infected for 1 hr with a gram-negative bacteria (Enteropathogenic *E. coli,* EPEC) or gram-positive bacteria (MRSA). Of note these two species were chosen due to their common extracellular preference in order to improve the isolation of bacteria undergoing Perforin-2 mediated bacteriolysis following infection with eukaryotic cells. In order to determine eukaryotic cellular contamination following bacterial purification, non-infected cells were processed in parallel as a control. Bacterial isolation and enrichment was successful as all fractions and filtrates were below detection for contaminating mammalian membranes and cellular debris as determined by immunoblot with antibodies targeting murine clathrin, actin, and GFP (data not shown).

Following the isolation procedure and subsequent differential rounds of centrifugation, Perforin-2 reactive bands with different molecular weights were detected in fractions containing either EPEC or MRSA. Perforin-2 was not detected after these fractions were passed through 0.22 μM filters (*Figure 6*). Prior to filtration each sample was spiked with goat IgG to account for nonspecific loss of soluble proteins. Unlike Perforin-2, goat IgG was equally detected in both bacterial fractions and filtrates. This indicates that Perforin-2 was not present as a soluble protein nor associated with cellular debris or microvesicles. Rather, Perforin-2 was associated with particles larger than 0.22 microns, such as bacterial cells. This conclusion was supported by the detection of the EPEC transcription factor ADA in the bacterial fractions but not filtrates. Likewise, MRSA Penicillin Binding Protein (PBP) was only detected in the unfiltered bacterial fractions.

Antibodies directed to the amino-terminal domains detected Perforin-2 in bacterial fractions. Two different banding patterns were observed depending upon which region the peptide derived Perforin-2 antibodies recognized (*Figure 6—figure supplement 1*). A single ca. 50 kD band was detected after probing with an antibody targeting amino-terminal portion of the Perforin-2 domain. Probing with either carboxy- or amino-terminal MACPF domain antibodies recognized two bands in

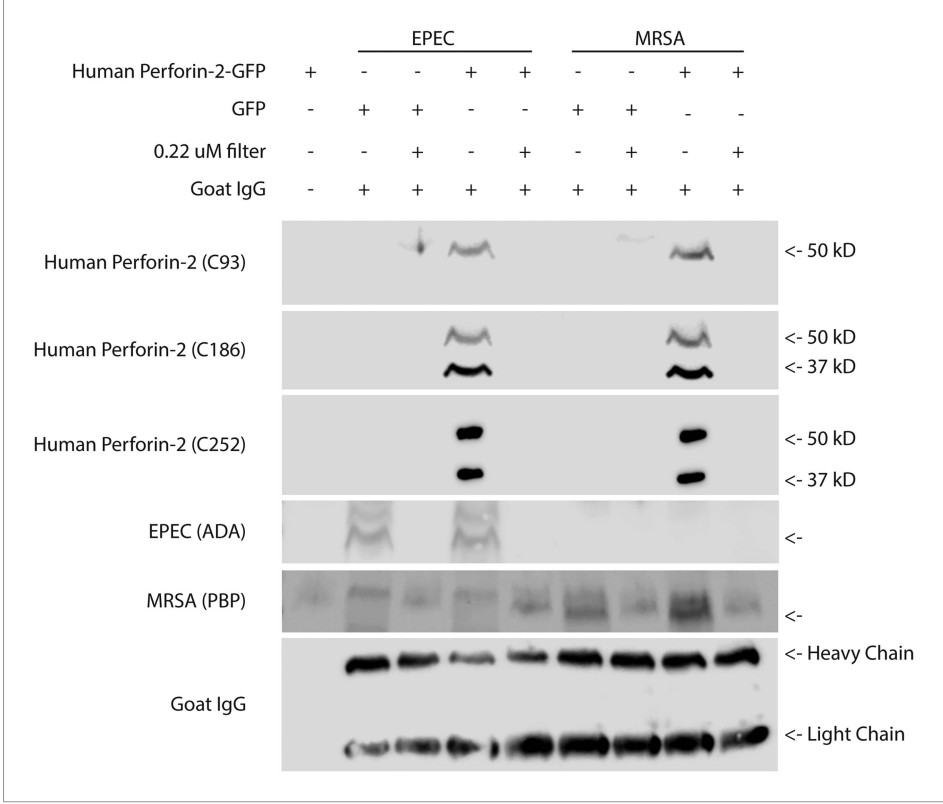

**Figure 6**. Cleaved Perforin-2 recovered after cellular infection with bacteria. Perforin-2 −/− MEFs were transfected with human Perforin-2-GFP or GFP induced with murine IFN-γ and infected with the extracellular bacteria MRSA or EPEC. Following infection, bacteria were isolated, goat IgG was added to assess for nonspecific protein loss, and a portion filtered to distinguish bacteria from debris/soluble proteins. A noninfected control (first lane), demonstrates the selectivity of the differential centrifugations to remove mammalian cells. *Figure 6—figure supplement 1* illustrates the recognition domain for human each Perforin-2 specific peptide generated antibody as well as verification. Fractions were probed against peptide-generated antibodies against the Perforin-2 domain (C93), and peptide generated antibodies against the MACPF domain (C186, C252). Commercial antibodies against ADA of EPEC and PBP of MRSA were utilized as bacterial markers. An additional band was observed following PBP immunoblot with a slightly higher molecular weight. This band was unspecific because it occurred in all samples including those derived from the experiments using EPEC. No signal was detected with previously validated peptide derived antibodies targeting the cytoplasmic domain of human Perforin-2 (C174), or peptide derived antibodies targeting a N-terminal portion of the Perforin-2 domain (C267) (Data not shown). In addition, commercial anti-human Perforin-2 antibody (detecting the cytoplasmic domain), clathrin, actin, and GFP also did not generate any signal (data not shown).

The following figure supplement is available for figure 6:

**Figure supplement 1**. Human Perforin-2 peptide antibody validation.

the bacterial fractions. One of these bands is the same size as the ca. 50kD band observed with the amino-terminal detecting Perforin-2 domain antibody, suggesting that this band is a combination of the MACPF domain and an amino-terminal portion of the Perforin-2 domain. Both MACPF domain antibodies also identified another band of ca. 37kD that is not detected with the Perforin-2 domain antibody, indicative of a MACPF domain fragment. Antibodies raised against the carboxy-terminal region of the Perforin-2 domain, the cytoplasmic domain, or the GFP tag were negative in unfiltered bacterial fractions despite validation of both Perforin-2 peptide antibodies with human macrophage whole cell lysates (data not shown, *Figure 6—figure supplement 1*). These results indicate that Perforin-2 makes physical contact with bacteria are supports are interpretation of electron dense poly-Perforin-2 pores.

## Perforin-2 deficient mice succumb to epicutaneous MRSA infection

We next sought to extend our in vitro studies to in vivo models using Perforin-2 deficient mice. Perforin-2 was neither vital for developmental processes nor for the control of commensal bacteria under specific pathogen-free conditions. This specifically included the regular development of the innate and adaptive immune system (*Figure 7—figure supplement 1*). This permitted us to study the antibacterial activity of Perforin-2 in vivo with a traditionally sub-lethal dose of MRSA and later other pathogens (*S. typhimurium*).

*S. aureus* is part of the normal bacterial skin flora in humans, but it can also become a major cause of serious skin and systemic infections. *S. aureus's* pathogenic arsenal, coupled with significant complement and antibiotic resistance, has allowed MRSA to evolve into a life-threatening, antibiotic-resistant pathogen in both community acquired as well as nosocomial settings. To determine the bactericidal role of Perforin-2 against MRSA in vivo, we utilized an epicutaneous MRSA challenge model for mice (*Cheng et al., 2009*; *Wanke et al., 2013*) in which hair removal is followed by 'tape-stripping' (*Wanke et al., 2013*) to disrupt the keratin barrier while exposing intact keratinocytes without overt wounding. *MPEG1* (Perforin-2) −/−, +/−, and +/+ littermates were challenged on the tape-stripped skin with $10^9$ CFU of MRSA, a PFGE type USA300 clinical isolate.

Perforin-2 knockout mice exhibited significantly decreased survival and more weight loss compared to Perforin-2 heterozygous or wild-type littermates (*Figure 7A,B*, *Figure 7—figure supplement 2*). To investigate the rate and route of the bacterial spread, 7 animals from each of the three genotypes were sacrificed 6 and 12 days following infection and the colony forming units (CFU) in their spleen, kidney, blood, and skin were determined (*Figure 7*, *Figure 7—figure supplement 3*). On day 6, all groups showed signs of systemic MRSA infection (recoverable CFU from internal organs). However, CFUs in Perforin-2 −/− mice were significantly higher than in wild-type mice (*Figure 7—figure supplement 3*) and by day 12, Perforin-2 deficient mice continued to have bacteremia with 100 to 100,000 fold higher MRSA counts in their organs as compared to Perforin-2 heterozygous or wild-type littermates (*Figure 7B–E*). The majority of heterozygous animals completely cleared the infection, with only a few mice demonstrating recoverable CFU in internal organs and the skin at day 12 (*Figure 7B–E*) while MRSA of the wild-type animals could only be recovered from the skin of three animals. On the other hand, all Perforin-2 knockout mice failed to eliminate MRSA and recover their weight. These animals eventually succumbed to infection (*Figure 7A*, *Figure 7—figure supplement 2*). Similar trends were observed with 129X1/SVJ congenic animals (*Figure 7B*) indicating that MRSA mortality was controlled by Perforin-2 and not attributable to confounding passenger mutations or genetic background differences between 129X1/SVJ and C57Bl/6 mice (*Vanden Berghe et al., 2015*).

We also investigated the in vitro control of MRSA by neutrophilic granulocytes because they contribute to abscess formation and, with some MRSA strains, are credited with the clearance of infection (*Rigby and DeLeo, 2012*). We found that MRSA replicated in Perforin-2 −/− neutrophils, but was killed in the presence of Perforin-2. Heterozygous neutrophils had intermediate bactericidal activity towards MRSA, suggesting that Perforin-2 was a rate-limiting molecule to control intracellular MRSA after infection (*Figure 7G*). These in vivo results indicate that Perforin-2 is required to limit the early systemic dissemination of MRSA (*Hahn et al., 2009*; *Onunkwo et al., 2010*) and, ultimately, to clear epicutaneous MRSA infection.

## Perforin-2 deficient mice succumb to *S. typhimurium* orogastric infection

To investigate whether the in vivo protection by Perforin-2 was limited to a particular site of infection or gram-positive pathogens, we also infected Perforin-2 deficient animals with gram-negative *S. typhimurium* via the orogastric route using well established protocols (*Barthel et al., 2003*). However, owing to the previously observed sensitivity of Perforin-2 deficient animals to MRSA, we decreased the infectious inoculum from the normal $LD_{50}$ of $10^8$ CFU–$10^5$ CFU.

As expected for this low infectious inoculum, Perforin-2 wild-type animals only transiently lost weight (<10%) whereas Perforin-2 deficient mice progressively lost weight and acquired severe diarrhea. Perforin-2 heterozygous littermates had more severe initial weight loss than wild-type animals, but these animals were able to recover (*Figure 8—figure supplement 1*). Equal *S. typhimurium* inoculation and colonization was confirmed by CFU analysis of the feces 12 hr following *S. typhimurium* inoculation (*Figure 8—figure supplement 2*). Finally, 129X1/SVJ congenic

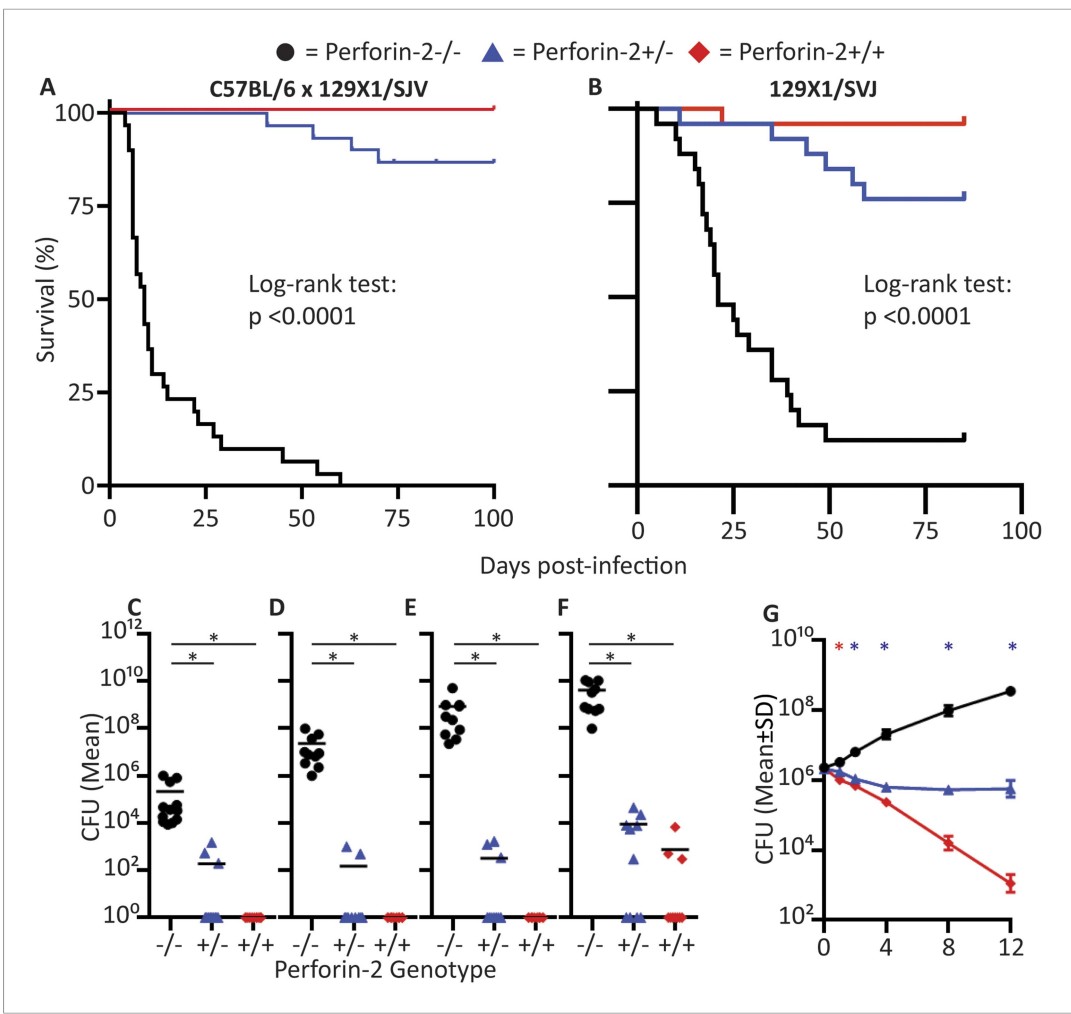

**Figure 7**. Perforin-2 is required for in vivo survival after MRSA epicutaneous challenge. (**A**) Aggregated survival curves of 60 C57BL/6 × 129 × 1/SJV mice challenged epicutaneously with $10^9$ MRSA. (**B**) Aggregated survival curves of 75,129X1/SVJ mice challenged epicutaneously with $10^9$ MRSA. (**C–F**) Organ load twelve days after MRSA epicutaneous infection in (**C**) blood, (**D**) spleen, (**E**) kidney, and (**F**) skin. (**G**) Perforin-2 *ex vivo* infection of murine neutrophils with MRSA. ◆= *MPEG1* (Perforin-2) wild-type animals (+/+), ▲= *MPEG1* (Perforin-2) heterozygous animals (+/−), ●= *MPEG1* (Perforin-2) knockout animals (−/−). Log-rank (Mantel–Cox) test was performed for A and B with statistical significance $p < 0.0001$. One-way ANOVA with Tukey post-hoc multiple comparisons was performed in C-G. *$p < 0.05$ as indicated. *$p < 0.05$ between Perforin-2 knockout:Perforin-2 wild-type and Perforin-2 knockout: Perforin-2 heterozygous neutrophils. *$p < 0.05$ between Perforin-2 knockout:Perforin-2 wild-type, Perforin-2 knockout:Perforin-2 heterozygous, and Perforin-2 heterozygous:Perforin-2 wild-type neutrophils.

The following figure supplements are available for figure 7:

**Figure supplement 1**. Characterization of lymphocytes in Perforin-2 knockout mice.

**Figure supplement 2**. Weight loss curves after MRSA epicutaneous infection.

**Figure supplement 3**. Epicutaneous MRSA infection Day 6 organ load.

---

animals were infected with $10^5$ *S. typhimurium* to address possibly confounding passenger mutations resulting from the different genetic background. As with the MRSA experiments, the observed Perforin-2 phenotype did not result from genetic differences because the aggregated survival curves were similar (compare *Figure 8B* with *Figure 8A*).

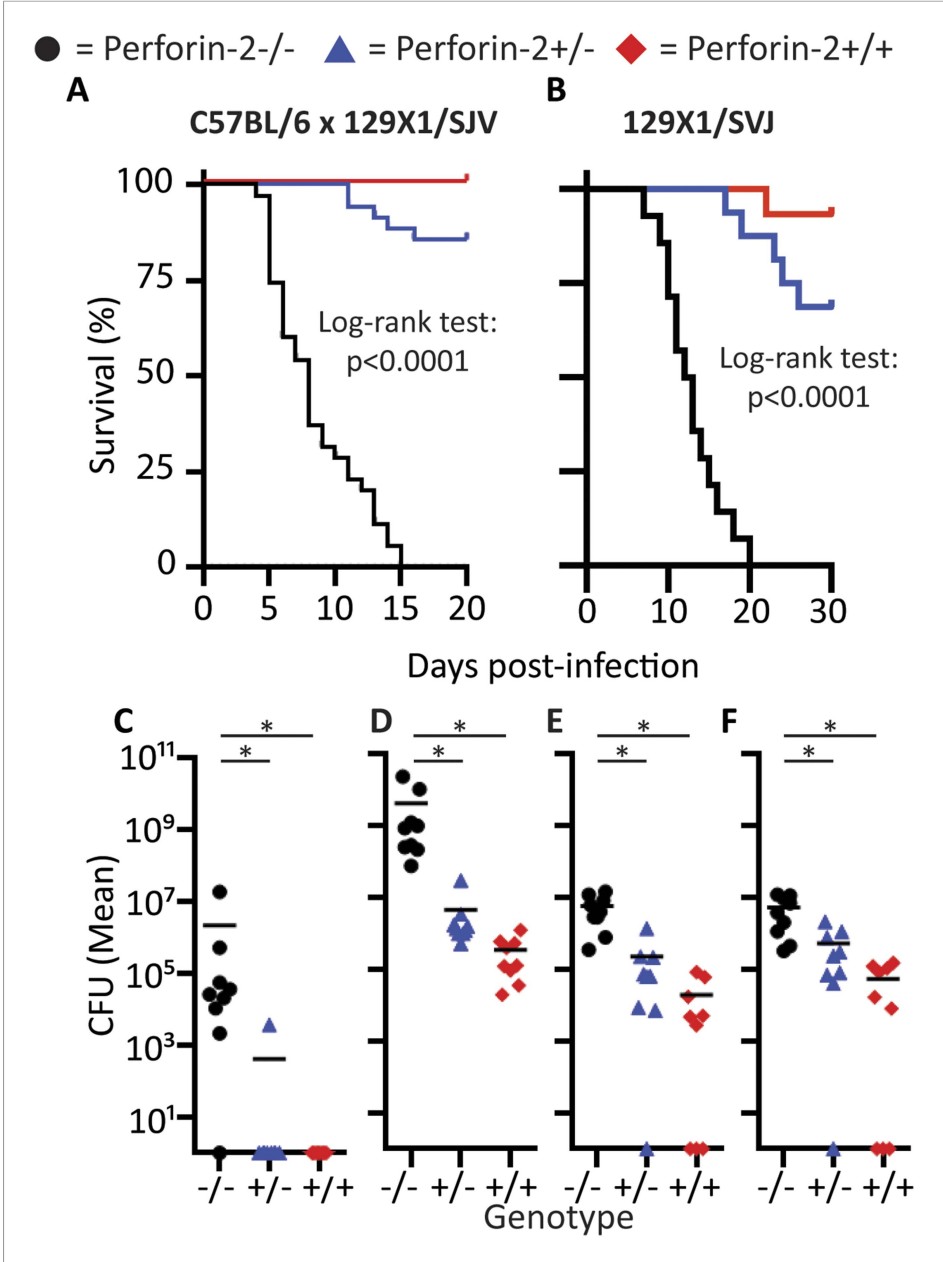

**Figure 8**. Perforin-2 is required for in vivo survival after orogastric *S. typhimurium* challenge. (**A**) Aggregated survival curves of 70 C57BL/6 × 129 × 1/SJV mice challenged with $10^5$ *S. typhimurium*. (**B**) Aggregated survival curves of 45,129X1/SVJ mice challenged with $10^5$ *S. typhimurium*. (**C–F**) Organ load five days after $10^5$ *S. typhimurium* infection in C57BL/6 × 129 × 1/SJV mice in (**C**) blood, (**D**) small intestine, (**E**) liver, and (**F**) spleen. ◆= *MPEG1* (Perforin-2) wild-type animals (+/+), ▲= *MPEG1* (Perforin-2) heterozygous animals (+/−), ●= *MPEG1* (Perforin-2) knockout animals (−/−). Log-rank (Mantel–Cox) test was performed for A and B with statistical significance p < 0.0001. One-way ANOVA with Tukey post-hoc multiple comparisons was performed in C-F. *p < 0.05 as indicated.

The following figure supplements are available for figure 8:

**Figure supplement 1**. Weight loss curves after in vivo orogastric *S. typhimurium* challenge.

**Figure supplement 2**. Equal Colonization in feces 12 hr following $10^5$ *S. typhimurium* inoculation.

**Figure supplement 3**. Organ load 60 hr after $10^5$ *S. typhimurium* oral-gastric infection.

Colonization of the animals with *S. typhimurium* was performed 60 hr following their infection (*Figure 8—figure supplement 3*). At this time point all three Perforin-2 groups showed signs of systemic *S. typhimurium* infection; however, with CFUs in Perforin-2 –/– mice were significantly higher than in wild-type animals. Moreover, only the former animals were septic with high bacterial titers in the blood. 5 days following infection, the bacterial burden of *S. typhimurium* in Perforin-2 –/– mice continued to be significantly higher in all organs as compared to their *MPEG1* (Perforin-2) +/+ littermates (*Figure 8C–F*) with intermediate levels observed in the heterozygous littermates.

These in vivo results indicate that Perforin-2 is also required to clear *S. typhimurium*, and other compensatory bactericidal mechanisms are unable to contain orogastric *S. typhimurium* contributing to the rapid systemic spread of the pathogen. All results together underscore the importance of Perforin-2 in the defeat of bacterial pathogens with individual cells cumulatively protecting entire organisms.

## Discussion

Much of our current understanding of antibacterial defenses is derived from studies of professional phagocytes, in particular their ability to migrate to sites of infection where they deploy antimicrobial effector molecules. Keratinocytes in the skin and the mucosal epithelia in the intestinal tract are known to provide local immune defense at barrier sites through the deployment of antimicrobial compounds and peptides to thwart pathogens while maintaining complex commensal communities. These effectors also, minimize chronic immune activation and inflammation (*Linden et al., 2008*; *Turner, 2009*). In this study we have characterized an additional effector, Perforin-2, present in professional phagocytes and barrier forming cells such as keratinocytes and mucosal epithelial cells. In the absence of Perforin-2, the ability of these cells to destroy bacteria is severely compromised (*Figure 2*, *Table 1*, *Table 2*). These findings suggest that Perforin-2 may act upstream of other antimicrobial effectors or that their potency is enhanced by Perforin-2.

Our study demonstrates that parenchymal, tissue-forming cells can also express Perforin-2 and its expression in these cells is essential for killing of bacteria. Our EM analysis of isolated bacteria suggests that this killing is achieved by polymerization and pore-formation of Perforin-2. In the absence of Perforin-2, the other innate defense effectors were unable to prevent the replication and systemic dissemination of bacterial pathogens. As per above, this suggests that Perforin-2 may facilitate or augment the activity of other antimicrobial effectors.

The unique structure of Perforin-2 as a membrane protein with a luminal and extracellular MACPF-killer domain is well suited for locally killing membrane-entrapped, extracellular or intracellular bacteria. Unlike Perforin-2, the other immunologic MACPF proteins do not act within the cell producing the effector protein. Perforin-1 is stored in granules in NK and CTL that are exocytosed into the immunologic synapse, thereby exposing Perforin-1 to extracellular calcium ions that trigger Perforin-1 polymerization in target membranes. The components of the MAC of complement are secreted by the liver into blood to exert their antibacterial effects after binding of C3b to the bacterial surface which initiates the assembly of the pore-forming C5b-8-poly-C9 complex.

Seemingly every cell can utilize Perforin-2 to defend itself against bacteria—potentially even before professional phagocytes are recruited–according to the constitutive expression of Perforin-2 in kc. Owing to this innate intrinsic protection of kc, bacteria may be cleared by these barrier cells even before the influx and assistance of professional phagocytes. The lack of barrier protection by kc when Perforin-2 is deficient may explain why Perforin-2 knockout mice rapidly (within days; *Figure 7—figure supplement 2*) exhibit symptoms to epicutaneous infection with MRSA. Thereafter the combined absence of Perforin-2 in the barrier tissue and in professional phagocytes may overwhelm the host by enhancing the systemic dissemination of the bacteria (*Figure 7—figure supplement 3*) and, ultimately, leading to multiple organ failure. The speed and severity of infections in Perforin-2 deficient animals, including sepsis and death following traditional non-lethal inocula of bacteria (*Figures 7, 8*), may also be assisted by the seemingly ubiquitous expression and function of Perforin-2, either inducible or constitutive. We suggest that this ubiquitous mechanism provides heretofore unrealized defenses at all potential sites of infection including barrier cells, phagocytes, parenchyma, and possibly even stroma. Further studies are required to determine the relative contribution of Perforin-2 expressed in each of them.

The scenarios above are further complicated by differences in the regulation of Perforin-2 in different cells. It is constitutively expressed in macrophages (and at least in certain barrier cells like kc) but requires induction in parenchymal cells. Therefore we speculate that unactivated parenchymal cells offer invasive bacteria a window of opportunity for replication and further dissemination. A second issue is that some pathogens have evolved mechanisms to suppress Perforin-2 (see accompanying report by McCormack et al.). On the other hand, our data also shows that when Perforin-2 is expressed at optimal levels, highly pathogenic and antibiotic-resistant bacteria can be killed by Perforin-2. This leads us to speculate that supraphysiologic levels of Perforin-2 expression could be an approach to defeat antibiotic-resistant bacteria. Vice versa, serious clinically relevant infections including antibiotic resistant strains of MRSA (*Figure 7*), *S. typhimurium* (*Figure 8*), and *M. avium* as well as *M. tuberculosis* (*Figure 1*) are all susceptible to Perforin-2, as demonstrated by greater bacterial pathogenesis in the absence of Perforin-2.

We undertook several approaches to determine whether Perforin-2 is a pore-forming protein and all approaches supported this concept. First, we isolated eukaryotic membranes from Perforin-2 overexpressing cells following limited trypsin digestion (to activate poly-Perforin-2 pore formation presumably by cleaving the cytoplasmic domain at a predicted trypsin-sensitive site) and observed 100 Å pores. The caveats of this approach entail its unphysiological setting especially the use of eukaryotic cell membranes that seem unlikely to compromise Perforin-2′s natural target. However, the visualization of pores on bacterial surfaces (*Figure 5*) was dependent on the availability of Perforin-2 and the bacterial membranes contained Perforin-2-derived fragments (*Figure 6*) that were only discernable following infection with Perforin-2 expressing cells. These results strongly imply that the antibacterial properties of Perforin-2 involve polymerization and pore-formation in bacteria. The detection of cleavage proteins suggests that Perforin-2 also could be cleaved physiologically at some point during the bacterial infection; however, the order of cleavage and pore-formation as well as the role of additional proteins requires further investigation.

Membrane perforation by the MAC of complement and by poly-Perforin-1 of cytolytic lymphocytes provides a conduit for additional effector molecules to ensure complete target destruction. MAC-polyC9 pores act as entry point for serum lysozyme to digest peptidoglycan resulting in bacterial collapse and lysis (*Schreiber et al., 1979*). Poly-Perforin-1 pores provide access to multiple granzymes that promote apoptosis of virally infected cells (*Dennert and Podack, 1983*; *Podack and Konigsberg, 1984*; *Masson and Tschopp, 1987*; *Jenne and Tschopp, 1988*; *Trapani et al., 1988*; *Shiver et al., 1992*; *Smyth et al., 1994*). Likewise, our studies suggest that poly-Perforin-2 pores enhance the delivery and action of endogenously produced ROS and RNS. This is consistent with our previous finding that exogenous lysozyme lysed *M. smegmatis* and MRSA when they were isolated from Perforin-2 expressing fibroblasts but failed to lyse these pathogens when they were isolated from Perforin-2 deficient fibroblasts (*McCormack et al., 2013*). Although high levels of lysozyme, ROS, or RNS can kill pathogens in the absence of Perforin-2 in vitro, we suggest that physiological endogenous levels of these–and potentially other antimicrobial effectors–may be limiting, requiring assistance by poly-Perforin-2 pore-formation to achieve bacterial killing (see *Figure 3* and its supplements).

As pathogenic microbes are specialized in their invasive strategies so are the innate strategies of the immune system in the use of the three pore forming proteins. First, interstitial extracellular bacterial pathogens are killed by MAC-polyC9 of the complement system whose components are present in the serum. Second, the viral production factories are destroyed by poly-Perforin-1 delivered by NK cells and CTL. Third, invasive or cell membrane attached bacterial pathogens are eliminated by poly-Perforin-2 expressed in most, if not all, cells. Common to all three pore-formers is that their antimicrobial activity cooperates with additional specialized effectors. Our studies add Perforin-2 as the third MACPF-domain driven pore-forming killer protein to the arsenal of the mammalian immune system for protection against microbial invasion.

Given the central role of Perforin-2 in antimicrobial responses, the pathogenicity of bacteria may depend on their ability to subvert or evade Perforin-2′s pore-forming ability. In the accompanying manuscript we describe one such virulence factor (Cif) that protects bacterial pathogens by blocking translocation of Perforin-2 (*McCormack et al., 2015*). Determining the mechanism of how other pathogenic bacteria inhibit Perforin-2 will elucidate additional critical steps in Perforin-2 activation and may allow the development of new treatments targeting both antibiotic-sensitive and antibiotic-resistant bacterial pathogens.

## Materials and methods

### Cell culture and bacterial organisms

RAW264.7 (TIB-71), J774A.1 (TIB-67), HL-60 (CCL-240), HeLa 229 (CCL 21), CATH.a (CRL-11179), Neuro-2A (CCL-131), NIH/3T3 (CRL-1658), Balb/c 3T3 (CCL-163), C2C12 (CRL-1772), CMT-93 (CCL-223), CT26.WT (CRL-2638), B16-F10 (CRL-6475), B16-F0 (CRL-6322), LL/2 (CRL-1642), MIA PaCa-2 (CRL-1420), Thp1 (TIB-202), NK-92 (CRL-2407), OVCAR3 (HTB-161), CaCo-2 (HTB-37), A549 (CCL-185), U-1752, JEG-3 (HTB-36), and HEK-293 (CRL-1573) cell lines were obtained from American Type Culture Collection, Manassas, VA. HUVECs were a gift from Dr. W Balkan, University of Miami, FL. BV2 microglial cell line was a gift from Dr. J Bethea, University of Miami, FL. ED-1 mouse lung adenocarcinoma cell line derived from the lung tumors of transgenic mice that express cyclin-E driven by the human Surfactant-C promoter was a gift from Dr. D Robbins, University of Miami, FL. MOVCAR 5009 and MOVCAR 5447 cells were a gift from Dr. D Connolly, Fox Chase Cancer Center, PA. A2EN primary-cell-like cervical epithelial cells were provided by Dr. K Fields, University of Kentucky, KY. UM-UC-3 and UM-UC-9 bladder cancer cell lines were a gift from Dr. B Grossman, MD Anderson Cancer Center, TX. Human adult and neonatal keratinocytes were obtained from Lonza. Primary murine astrocytes and primary CNS fibroblasts were a gift from Dr. J Lee, University of Miami, FL. Neonatal ventricular myocytes were a gift from Dr. N Bishopric, University of Miami, FL.

All cells were cultured at 37°C in a humidified atmosphere containing 5% $CO_2$ following ATCC recommendations for culture conditions. HL-60 were differentiated to neutrophils using retinoic acid or to macrophages using PMA as previously described (*Meyer and Kleinschnitz, 1990*; *Daigneault et al., 2010*). Murine primary macrophages were obtained from thioglycolate-elicited peritoneal or differentiated from bone marrow utilizing M-CSF as previously described (*Zhang et al., 2008*). Murine bone marrow derived dendritic cells were differentiated from bone marrow utilizing GM-CSF (*Dearman et al., 2009*). Human monocyte derived macrophages and human monocyte derived dendritic cells were differentiated from monocytes as described previously (*Vijayan, 2012*) and human PMN isolated as previously described (*Oh et al., 2008*). Primary human NK cells were isolated utilizing RosetteSep Human NK cell Enrichment cocktail from Stemcell Technologies. Murine embryonic fibroblasts (MEFs) and murine PMN were isolated as previously described (*Luo and Dorf, 2001*; *Scheuner et al., 2001*).

*S. typhimurium* LT2 (ATCC 700720) and SL1344 (gift from Dr. J. Galán, Yale University), and *E. coli* K12 DH5α were grown in Luria–Bertani broth (LB) or heart infusion broth (HIB; Becton, Dickinson and Co., Sparks, MD, United States) at 37°C. *Staphylococcus aureus* CLP148 and CLP153 (MRSA PFGE type USA300) were grown in LB or tryptic soy broth (Sigma–Aldrich, St. Loius, MO, United States) at 37°C. *M. avium* (gift from Dr. T. Cleary, University of Miami), and *M. smegmatis* (ATCC 700084) were grown in Middlebrook 7H9 broth. *S. typhimurium* LT2 carrying deletions in *hmpA* (codons 35 to 361) and *sodC1* (codons 16 to 142) were generated via lambda Red-mediated recombination (*Datsenko and Wanner, 2000*) with modifications as described by Bartra et al (*Bartra et al., 2008*), and cultured in the same manner as wild-type *S. typhimurium*.

### Generation of Perforin-2 knockout mice

For the generation of Perforin-2 knockout mice the targeting vector was linearized and electroporated into RW-4 ES cells originating from the 129X1/SvJ strain, followed by selection in G418. Targeted clones were screened by PCR. From 90 clones, 2 positive clones were selected that had undergone homologous recombination and were identified through Southern blot analysis. One ES clone was utilized for the generation of chimeric mice by injection using C57Bl/6J blastocysts as the host. The resulting female chimeras were further mated with C57Bl/6J male mice for germ line transmission. The heterozygous mice ($F_1$ mice) were interbred to obtain wild-type, heterozygous, and homozygous littermates ($F_2$). C57Bl/6 × 129 × 1/SvJ animals utilized in these experiments were backcrossed 8–10 times for these experiments. Mouse genotype was determined by PCR using PCR probes MP10 and MP11.

To generate 129X1/SvJ inbred animals without potential ES cell passenger mutations, chimeric mice were mated with 129X1/SvJ animals, to assess for germ line transmission. The heterozygous mice were then interbred to obtain a genetically pure 129X1/SvJ strain. Mouse genotype was determined by PCR utilizing PCR probes MP10 and MP11.

Animals were bred at the University of Miami, Miller School of Medicine Transgenic Core Facility. Mice were allowed to freely access food and water and were housed at an ambient temperature of 23°C on a 12 hr light/dark cycle under specific pathogen-free condition. Animal care and handling were performed as per IACUC guidelines.

## Methicillin resistant *S. aureus* in vivo epicutaneous infection

All animal experiments were performed in accordance with University of Miami Animal Care and Use Committee guidelines. Animal's genotype was blinded prior to the experiment to limit bias. Our methodology was adopted from references (*Cheng et al., 2009*; *Wanke et al., 2013*). In brief, all mice were shaved and tape-stripped (7 applications) with Transpore tape (3M, Minneapolis, MN, United States). This level of tape stripping did not create a wound, but was sufficient to disrupt the epidermal barrier. An inoculum of $10^9$ MRSA strain CLP 153 in 0.02 ml of phosphate-buffered saline (PBS) or PBS control was added to ~1 cm$^2$ of skin and the area bandaged with plastic sheet and overwrapped with dressings of Transpore tape and Nexcare waterproof tape (3M) for 6 hr, at which time the bandage was removed. Mice were weighed daily throughout the experiment; animals were euthanized after greater than 30% weight loss.

For CFU enumeration, mice were sacrificed either 6 or 12 days after infection, cardiac puncture was performed and organs were harvested, weighed, and homogenized using a potter homogenizer in ddH$_2$O with 0.05% Triton X-100. The homogenates were diluted and plated on TSA II plates (kanamycin and oxacillin selection). All samples were normalized based on weight.

## S. typhimurium in vivo infection

All animal experiments were performed in accordance with University of Miami Animal Care and Use Committee guidelines. Animal's genotype was blinded prior to the experiment to limit bias. Mice were infected orogastrically with $10^5$ colony-forming units of Streptomycin resistant *S. typhimurium* (SL1344) 24 hr after orogastric Streptomycin pretreatment as previously described (*Barthel et al., 2003*). Mice were weighed daily throughout the experiment; the animals were euthanized after greater than 30% weight loss.

For CFU enumeration, mice were sacrificed 3 days after $10^5$ *S. typhimurium* orogastric infection, cardiac puncture was performed and organs were harvested, weighed, and homogenized using a potter homogenizer in ddH$_2$O with 0.05% Triton X-100. The homogenates were diluted and plated on MacConkey agar plates (streptomycin at 100 µg/ml). All samples were normalized based on weight.

## M. tuberculosis fluorescent counts

Mouse bone marrow-derived macrophages were cultured in DMEM containing 15% L929-conditioned medium, 10% fetal bovine serum, and 2 mM glutamate for 7 days. Macrophages were plated in 96 well plates and infected at an MOI of 3:1 with *M. tuberculosis* CDC 1551 expressing the fluorescent protein mCherry under constitutive promoter (*smyc'::mCherry*). Bacterial survival and growth was monitored by measuring mCherry fluorescence using a Perkin–Elmer EnVision plate reader. This assay has been previously validated by comparison with colony forming unit (CFU) counts (*Lee et al., 2013*).

## Immunoblot

Anti-murine Mpeg1 (ab25146), anti-human Mpeg1 (ab176974), anti-calreticulin (ab22683), anti-GM130 (ab52649), anti-clathrin (ab2731), anti-Methicillin Resistant *Staphylococcus Aureus* (ab73263), anti-Ada (ab18104), anti-pan Cadherin (ab140338), and anti-catalase (ab16731) (Abcam, Cambridge, MA); anti-GFP (sc9996), and anti-EEA1 (sc-6415) (Santa Cruz Biotechnology, Dallas, TX); anti-Prohibitin (Poly6031), anti-Lamp1 (1D4B), and anti-β-actin (2F1-1) (Biolegend, San Diego, CA, United States); anti-Golgin-97 (Thermo Fisher Scientific, Waltham, MA, United States); MEK1/2 (D1A5) (Cell Signaling Technology, Danvers, MA, United States); peptide synthesized cytoplasmic Perforin-2 antibody (21$^{st}$ Century Biochemicals, Marlborough, MA, United States); and peptide synthesized human Perforin-2 antibodies against each domain (Abmart, Berkeley Heights, NJ, United States) were utilized for immunoblots as indicated.

Densitometry analysis was performed where indicated utilizing ImageJ software.

## Flow cytometry and antibodies

Antibodies directed against murine CD3 (clone 145-2C11, Biolegend), CD4 (clone RM4-5, Biolegend), CD8 (clone 53-6.7, Biolegend), NK1.1 (clone PK136, Biolegend), CD62L (clone MEL-14, Biolegend), CD44 (clone IM7, Biolegend), and CD19 (clone 6D5, Biolegend) were used in multicolor FACS analysis. Samples were washed and resuspended in cold flow cytometry staining buffer (1% BSA/PBS); stained for 30 min in the dark before a final wash and acquisition. All samples were acquired on a BD Fortessa Flow cytometer running FACS DIVA software. Analysis was performed using FlowJo X software (TreeStar; OR, United States).

## Subcellular fractionation of endogenous perforin-2

Subcellular fractionation was performed on human THP-1 cell line induced with PMA for 16 hr and then allowed to rest without PMA stimulation for 48 hr. Samples were isolated following Axis-Shield Density Gradient methods (Axis-Shield, Norway) for exocytosis analysis: resolution of plasma membrane from TGN/endosomes and cytosolic proteins in self-generated gradient (S45). In brief, cells were harvested and homogenized utilizing a Dounce homogenizer. The homogenate was then centrifuged to pellet nuclei and other cell debris. The post-nuclear supernatant was then loaded equally on decreasing concentrations of iodixanol (30%, 20%, 10%) and centrifuged at 353,000g for 3 hr. The gradient was then collected in 0.1 ml fractions by aspiration from the meniscus.

For better resolution of clathrin-coated vesicles, endosomes, and lysosomes protocol S43 was utilized. In brief, cells were homogenized utilizing a Dounce homogenizer, and centrifuged to remove nuclei and other cell debris. The supernatant was then mixed with 12.5% iodixanol underlaid with 20% iodixanol. The mixed and layered tube was then centrifuged at 350,000g for 1.5 hr. The gradient was collected in 0.1 ml fractions by aspiration from the meniscus. For better resolution of peroxisomes, protocol S13 was utilized—Purification of mammalian peroxisomes in a self-generated gradient.

After collection, fractions were analyzed for protein content by DC Protein Assay (Bio-Rad, Hercules, CA, United States), and screened for localization of subcellular fractions by Western blot analysis.

## Intracellular bacterial load

The intracellular gentamicin protection assays were conducted as previously described (*Lutwyche et al., 1998*; *Laroux et al., 2005*; *McCormack et al., 2013*). Briefly, 100 ng/ml of species specific human or murine IFN-γ was added 14 hr before infection where necessary to uniformly induce Perforin-2 expression. In all graphs bacteria were added as indicated, and after 30–60 min to allow for uptake/invasion, the extracellular bacteria were washed and re-plated in gentamicin supplemented medium. For gentamicin protection assays, the multiplicity of infection was between 20 and 50 bacteria per cell to allow for sufficient uptake of bacteria.

For gentamicin-free intracellular assessment of bacterial load, the gentamicin protection assay was modified as follows: achieve >90% eukaryotic cell confluence on infection, decrease multiplicity of infection from between 20–50 to between 0.5–5, and trypsinize eukaryotic cells after initial wash steps to help eliminate attached extracellular bacteria. Every 4 hr, the medium was removed and checked for extracellular bacterial growth, washed twice with PBS, and replaced with fresh medium.

## Isolation of extracellular bacteria for perforin-2 fragments

The gentamicin-free infection assay described above was modified as follows to increase extracellular bacterial recovery: utilization of pathogens that do not facilitate uptake (enteropathogenic *E. coli* and MRSA), increased inoculation to a MOI from 0.5–5 to 10–30, and decreased invasion/attachment time from 60 min to 40 min. After the attachment/early infection phase was complete, cells were washed 5x with prewarmed media to remove non-adherent bacteria, and infection was allowed to continue for an additional 30 min. After this was complete, cells were washed an additional 3x with PBS and trypsinized to remove both cells as well as extracellular bacteria.

Enrichment of extracellular bacteria was performed by successive low-speed differential centrifugations to remove intact mammalian cells from extracellular bacteria. Cell/bacteria from above were spun at 200g for 15′ with collection of the supernatant, the supernatant was collected and respun for a total for 7 spins. Non-infected mammalian cells were also treated in the same fashion to quantify for mammalian cellular contamination. Following the last spin from above, the supernatant

was spun at 20,000g for 15′ to pellet all bacteria and cellular debris. Goat IgG was added to the bacterial enriched pellet following resuspension and part of the post-infection bacterial/goat IgG solution was then filtered through prewashed 0.22 μm filters. After filtration, the eluent was collected and fractions were mixed and boiled with reducing laemmli sample buffer and loaded on SDS-PAGE for Immunoblotting.

## RNA interference

For murine cells, RNA interference and transfection were conducted as previously described (*McCormack et al., 2013*). For human cells, the aforementioned murine system was modified through utilizing three human Perforin-2-specific silencer select siRNAs purchased from Thermo Fisher Scientific Silencer Select #s61053, s47810, s61054. Silencer select negative control #1 and 2 from) Thermo Fisher Scientific were also utilized as a negative control.

## Quantitation of gene expression via qRT-PCR

Murine or human RNA extraction, cDNA synthesis and analysis was performed as previously described (*Fields et al., 2013*; *McCormack et al., 2013*). Message for the housekeeping gene GAPDH was amplified as an internal normalization control. All assays were performed in technical triplicate for each RNA sample.

## Assessment of nitrite and ROS detection

For nitrite detection, adherent PEM were stimulated overnight with IFN-γ (100 ng/ml) and stimulated with LPS (100 ng/ml) in the presence of N-Acetyl Cysteine (NAC) (Sigma-Aldrich, St. Louis, MO) or $N^G$-nitro-L-arginine methyl ester (L-NAME) (Sigma-Aldrich) for 48 hr. 50 μl of supernatant was collected for analysis of $NO_2-$ production using Griess reagents. 50 μl of 1% sulfanilamide in 3% $H_3PO_4$ was added to 50 μl of supernatant followed by 50 μl of 0.1% napthylethylene dihydrochloride in 3% $H_3PO_4$ and the wells were read on a spectrophotometer at 550 nM. Sodium nitrite was used a standard at concentrations ranging from 1 μM to 125 μM.

For ROS detection, adherent PEM were stimulated overnight with IFN-γ (100 ng/ml) and then labeled with 10 μM CM-$H_2$DCFDA (Thermo Fisher Scientific) in PBS for 30 min at 37˚C, followed by washing and addition of complete media. Inhibitors were added 30 min prior to addition of LPS, PMA (1 μM), or $H_2O_2$ (200 μM). 30 minutes later, cells were scraped and immediately analyzed by flow cytometry.

## Confocal imaging

For live cell imaging, RAW264.7 or BV-2 cells were nucleofected with Perforin-2-GFP and stimulated overnight with LPS (1 ng/ml) and IFN-γ (100U/ml) in glass bottom dishes with No. 1.5 coverglass (MatTek Corp, Ashland, MA, United States). Cells were washed once with PBS and organelles were labeled. For endoplasmic reticulum (ER) labeling, ER-Tracker™ Blue-White DPX (Thermo Fisher Scientific) was used at a working concentration of 1 μM for 30 min at 37˚C. For all other stains, transfected cells were fixed with 3% paraformaldehyde (PFA) for 15 min at room temperature, permeabilized with 0.5% saponin, blocked with 10% normal goat serum and incubated with primary and secondary antibodies. Anti-CD107a (LAMP-1) (BD Pharmingen, San Jose, CA, United States), anti-EEA1 (EMD Millipore), and anti-GM130 (BD Biosciences) were used to identify cellular organelles. Secondary antibodies were all raised in goats. Images were taken on a Leica SP5 inverted confocal microscope with a motorized stage and analyzed using Leica application suite advanced fluorescence software.

## Electron microscopy

### Eukaryotic cell membranes

Membranes were isolated from stably transfected Perforin-2-GFP HEK-293 cells by $N_2$-cavitation and differential centrifugation. Membranes were resuspended in a small volume of neutral Tris-buffered saline, treated with 100 μg/ml trypsin for 1 hr at 37˚C, washed and negatively stained with 5% neutral Na-phosphotungstate for 30 s. Images were taken at 52,000-fold initial magnification on a Phillips CM10 transmission electron microscope.

## Bacterial membranes

MEF were stimulated for 14 hr with IFN-γ (100 U/ml) and infected with the indicated bacterial strains at a multiplicity of infection of 30 for 5 hr. Prokaryote membranes were harvested through lysing MEFs with the non-ionic, non-denaturing detergent 1% Igepal in ddH$_2$O. The lysate was centrifuged at 200g for 10 min to pellet intact bacteria; intact bacteria were subsequently sheared with a polytron to disrupt intact bacteria and separate the bacterial surface (cell wall and membrane). The resulting pellet was treated with 100 μg/ml trypsin for 1 hr at 37°C, sedimented and resuspended in minimal ddH$_2$O and negatively stained with 3% uranyl formate for 30 s. Images were taken between 52,000 to 168,000-fold initial magnification on a Phillips CM10 transmission electron microscope.

## Statistical analysis

Students t-test, multiple t-test with Holm-Sidak multiple comparisons correction, one-way ANOVA with Bonferroni multiple comparisons test, or Kruskal–Wallis non-parametric test with Dunn's multiple comparison test was used for comparisons (GraphPad Prism Version 6.0b and SPSS 21.0 were utilized for statistical analysis).

## Acknowledgements

We acknowledge the help of Dr. Y Wang in generating Perforin-2 −/− mice and the advice and assistance of Dr. V Deyev, Dr. B Watson, Dr. Sara Schesser Bartra, and E Fisher, University of Miami, FL. We would like to thank Drs. J Bethea, L Plano, T Cleary, W Balkan, J Lee University of Miami, FL, Dr. D Connolly, Fox Chase, PA, Dr. B Grossman, MD Anderson, TX, Dr. K Fields University of Kentucky, KY, and Dr. J Galan, Yale University, CT, for providing mammalian cell lines and bacterial strains. We are indebted to Dr. M Bartlett Bunge and P Bates, University of Miami, for help in obtaining the electron microscopic images, and Dr. G McNamara for assistance in obtaining the confocal images.

## Additional information

### Competing interests

RMMC: This author is an inventor of patents used in the study and stand to gain royalties from future commercialization. LRA: This author is an inventor of patents used in the study and stand to gain royalties from future commercialization. KL: This author is an inventor of patents used in the study and stand to gain royalties from future commercialization. ERP: This author is an inventor of patents used in the study and stand to gain royalties from future commercialization. The other authors declare that no competing interests exist.

### Funding

| Funder | Grant reference | Author |
|---|---|---|
| National Institutes of Health (NIH) | CA039201 | Eckhard R Podack |
| Lois Pope Life Foundation | Developmental Fellowship | Ryan M McCormack |
| National Institutes of Health (NIH) | CA109094 | Eckhard R Podack |
| National Institutes of Health (NIH) | AI0073234 | Eckhard R Podack |
| National Institutes of Health (NIH) | AI096396 | Eckhard R Podack |
| National Institutes of Health (NIH) | DK0942601 | Robert S Kirsner |
| National Institutes of Health (NIH) | NR013881 | Marjana Tomic-Canic |
| National Institutes of Health (NIH) | AI106290 | Ryan M McCormack |
| National Institutes of Health (NIH) | AI110810 | George P Munson, Eckhard R Podack |

| Funder | Grant reference | Author |
| --- | --- | --- |
| National Institutes of Health (NIH) | AI107062 | Marjana Tomic-Canic, Robert S Kirsner, Eckhard R Podack |
| National Institute of Nursing Research (NINR) | NR015649 | Marjana Tomic-Canic, Eckhard R Podack |

The funders had no role in study design, data collection and interpretation, or the decision to submit the work for publication.

## Author contributions

RMMC, MS, ERP, Conception and design, Acquisition of data, Analysis and interpretation of data, Drafting or revising the article, Contributed unpublished essential data or reagents; LRA, DGF, MLO, MGL, AM, KL, LEG, NS, NS, OS, GVP, GPM, MT-C, RSK, DGR, Conception and design, Acquisition of data, Analysis and interpretation of data, Drafting or revising the article

## Author ORCIDs

George P Munson, http://orcid.org/0000-0002-3692-8199

## Ethics

Animal experimentation: This study was performed in strict accordance with the recommendations in the Guide for the Care and Use of Laboratory Animals of the National Institutes of Health. All of the animals were handled according to approved institutional animal care and use committee (IACUC) protocols (#13–233 and #12–259) of the University of Miami Miller School of Medicine.

# Additional files

## Supplementary files

• Supplementary file 1. Type I and Type II interferon increase Perforin-2 message in murine non-hematopoietic cell lines. (A–D) Select murine cell lines from *Table 1* indicating qPCR delta CT (Perforin-2 normalized to GAPDH) (five experimental replicates) following Type I (Interferon-α, β stimulation), Type II (Interferon-γ stimulation), or both Type I and II (Interferon-αβγ stimulation). (A) Ovarian cancer cell line MOVCAR 5009, (B) Cath.a neuroblastoma cell line, (C) C2C12 myoblast cell line, (D) B16-F10 melanoma cell line. (E–H) Interferon stimulation corresponds with an increase in Perforin-2 protein. (E) Ovarian Cancer MOVCAR 5009 and (F) C2C12 myoblast cell line. Densitometry analysis of five experimental replicates of (G) MOVCAR 5009 or (H) C2C12. (A–D) Statistical analysis was performed with one-way ANOVA with Tukey post-hoc multiple comparisons. (G, H) Statistical analysis was performed with Student's T-test. *p < 0.05.

• Supplementary file 2. Type I and Type II interferon increase Perforin-2 message in human non-hematopoietic cell lines. Select human cell lines from *Table 2* analyzed by qPCR demonstrating delta CT (Perforin-2 normalized to GAPDH) (five experimental replicates) after Type I (Interferon-αβ stimulation), Type II (Interferon-γ stimulation), or both Type I and II (Interferon-αβγ stimulation). (A) Primary HUVEC cells, (B) HEK293 cell line, and (C) MIA-PaCa-2 pancreatic cancer cell line. Interferon stimulation also increased human Perforin-2 protein with (D) MIA-PaCa-2 and (E) HUVEC cell lines. Densitometry analysis of five experimental replicates of (F) MIA-PaCa-2 or (G) HUVEC. (A–C) Statistical analysis was performed with one-way ANOVA with Tukey post-hoc multiple comparisons. (F, G) Statistical analysis was performed with Student's T-test. *p < 0.05.

• Supplementary file 3. Perforin-2 significantly contributes to intracellular killing in murine non-hematopoietically derived cells. (A–C) One day prior to the experiment, cells were transfected with either a pool of scramble (□) or murine Perforin-2 specific (■) siRNA and 14 hr prior to the experiment induced with IFN-γ. (A) MOVCAR 5009 infected with *S. typhimurium*, (B) CT26 infected with MRSA, or (C) C2C12 infected with *M. smegmatis*. The above graphs contain 5 biologic replicates, and are representative of 3 independent experiments. Statistical analysis was performed utilizing multiple T-tests with correction for multiple comparisons using the Holm-Sidak method. *p < 0.05.

• Supplementary file 4. Perforin-2 siRNA knockdown is efficient in both murine and human cells. (A–D) Representative blots from knockdown of selected cells from *Table 1*, *Table 2*, and *Figure 3*. (A, C) Human HUVEC cells representing knockdown of human Perforin-2; (B, D) Murine C2C12 myoblast cell line-demonstrating knockdown of murine Perforin-2. Cells were transfected with either a pool of human or mouse Perforin-2 specific siRNA or scramble siRNA. Cells were induced for 14 hr with IFN-γ and 24 hr post-transfection lysed for protein quantification. Statistical analysis was conducted utilizing Student's T-test. *$p < 0.05$.

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
