## [Decision Letter]

Thank you for sending your work entitled “Perforin-2 is essential for intracellular defense of parenchymal cells and phagocytes against pathogenic bacteria” for consideration at *eLife*. Your article has been favorably evaluated by Richard Losick (Senior Editor), a Reviewing Editor, and two reviewers.

The Reviewing editor and the reviewers discussed their comments before we reached this decision, and the Reviewing editor has assembled the following comments to help you prepare a revised submission.

We agree that your manuscript reports an important finding but that there is a need to reinforce the results and/or to modulate your claims. Here are the main criticisms:

1) The authors show clearly that Perforin-2 promotes in vivo defense against methicillin resistant *S. aureus* (MRSA) as demonstrated by greatly increased susceptibility of Perforin-2*–/–* mice. However, demonstration of similar in vivo protection using another bacterial pathogen would enhance the study.

2) One serious problem with the paper is that the mice were not backcrossed into the C57BL background for multiple generations. Conclusions about infection experiments in mice are repeatedly found to be unreliable if this isn't done.

3) It would be good to know whether Perforin-2 is more abundant in certain cells and, when induced, how levels compare (impossible to evaluate fold increase over “zero”).

4) How does Perforin-2 work on both gram-positive and gram-negative bacteria? They have very different cell wall. How does Perforin-2 differentiate between bacterial and mammalian cell membranes? The so-called pores in bacterial membranes are not completely convincing. No evidence is provided to support the notion that these pores are formed by polymerized Perforin-2, as stated – this is just an inference. Could immunogold staining help localize Perforin-2 to the “pores”? Is an endogenous protease needed? It would be helpful to know if Perforin-2 has a lower MW or separates from the host membrane and relocalizes to the bacterial membrane after bacterial infection. These points require clarification.

5) The authors argue that ROS and NOS enhance the bactericidal activity of Perforin-2. However, all experiments have been with chemical inhibitors (NAC and L-NAME) and follow-up analysis with knockout mice must be done to really demonstrate this conclusion or else the language should be tempered.

6) The authors claim that Perforin-2 is not required for normal murine development or for control of commensal bacteria. Are there data to support this statement or literature to cite?

7) In the Discussion, the authors draw sharp distinctions among extracellular bacterial pathogens, intracellular viral pathogens and intracellular bacterial pathogens and their use of MAC-polyC9, poly-Perforin-1 and poly-Perforin-2. However, the experiments using Perforin-2*–/–* mice use *S. aureus*, which is primarily an extracellular pathogen although it can invade some cell types. This discussion should be tempered to soften the sharp distinctions that are made.

The Discussion talks about 3 immune cell pore-forming proteins (complement, Perforin and Perforin-2). Why is GNLY not mentioned, especially since it has anti-bacterial activity? It would be good to relate the findings here to the recent paper of Walch et al. in Cell.

---

## [Author Response]

[…] We agree that your manuscript reports an important finding but that there is a need to reinforce the results and/or to modulate your claims. Here are the main criticisms:

*1) The authors show clearly that Perforin-2 promotes in vivo defense against methicillin resistant* S. aureus *(MRSA) as demonstrated by greatly increased susceptibility of Perforin-f2–/– mice. However, demonstration of similar in vivo protection using another bacterial pathogen would enhance the study.*

We agree. Thus we have added orogastric infection with the gram-negative pathogen *S. typhimurium*. As in previous experiments, similar increased susceptibility of Perforin-2 knockout mice was observed (Figure 8, Figure 8—figure supplement 1, Figure 8—figure supplement 2 and Figure 8—figure supplement 3).

2) One serious problem with the paper is that the mice were not backcrossed into the C57BL background for multiple generations. Conclusions about infection experiments in mice are repeatedly found to be unreliable if this isn't done.

In our initial study the Perforin-2 mutation was backcrossed 7-10 times, a point that is now made clearer (subsection “Generation of Perforin-2 knockout mice”). However, we agree that is not sufficient to remove doubts as per the cause of the phenotype. Therefore we now include additional animal studies with genetically pure 129X1/SvJ mice (Figures 7 and 8). The results with these new mice are similar to those previously reported (Figures 7 and 8). Therefore we are confident that the reported phenotypes are due to Perforin-2 deficiency and not the result of passenger mutations or differences between strains.

3) It would be good to know whether Perforin-2 is more abundant in certain cells and, when induced, how levels compare (impossible to evaluate fold increase over “zero”).

We agree and have modified our qPCR analysis to demonstrate induction by ΔCT of Perforin-2 subtracted from our housekeeping gene (GAPDH). With this analysis one can now compare Perforin-2 levels between samples pre- and post-stimulation as well as between samples.

4) How does Perforin-2 work on both gram-positive and gram-negative bacteria? They have very different cell wall. How does Perforin-2 differentiate between bacterial and mammalian cell membranes? The so-called pores in bacterial membranes are not completely convincing. No evidence is provided to support the notion that these pores are formed by polymerized Perforin-2, as stated – this is just an inference. Could immunogold staining help localize Perforin-2 to the “pores”? Is an endogenous protease needed? It would be helpful to know if Perforin-2 has a lower MW or separates from the host membrane and relocalizes to the bacterial membrane after bacterial infection. These points require clarification.

This is an interesting mechanistic question regarding pore-formation of Perforin-2 on both gram-positive and gram-negative bacteria as well as targeting differentiation between bacterial or mammalian cell membranes. However, based on previous studies with other MACPF proteins in complement as well as Perforin-1, both of these questions are likely to require a significant amount of work that goes beyond the scope of this current study.

We agree that immunogold staining would help to localize Perforin-2 to the visualized pores. Unfortunately no antibodies are currently available that would work with immunogold staining of Perforin-2. In order to clarify that the visible pores are poly-Perforin-2, we have added blinded quantification of pores present with different conditions (Figure 5—figure supplement 1). We have also isolated both gram-positive and –negative bacteria following infection and have identified Perforin-2 fragments present with the bacteria. This finding suggests that Perforin-2 can be isolated in both gram-positive and -negative bacteria following infection and that poly-Perforin-2 may be what is observed with the EM.

5) The authors argue that ROS and NOS enhance the bactericidal activity of Perforin-2. However, all experiments have been with chemical inhibitors (NAC and L-NAME) and follow-up analysis with knockout mice must be done to really demonstrate this conclusion or else the language should be tempered.

We have tempered the claims regarding Perforin-2 with ROS and NOS. Perforin-2 and ROS or NOS knockout animals are not currently available, a clarification that we believed was necessary (subsection “Reactive oxygen and nitrogen species enhance the bactericidal activity of Perforin-2 but have little microbicidal activity when Perforin-2 is absent”). We have also clarified the rational for using mutant bacteria that is attenuated to ROS and NOS.

6) The authors claim that Perforin-2 is not required for normal murine development or for control of commensal bacteria. Are there data to support this statement or literature to cite?

In Figure 7—figure supplement 1, we analyze the frequency of CD4, CD8, B cell, and NK cells in the peripheral blood of both Perforin-2*–*/*–* and +/+ littermates. As these Perforin-2 knockout animals are able to thrive in specific pathogen free barrier conditions, we conclude that Perforin-2 is not required for normal murine development or for control of commensals. To the extent of our knowledge, this is the first generation of Perforin-2 knockout mice. Therefore, we are unable to cite other literature confirming the normal development of these animals.

*7) In the Discussion, the authors draw sharp distinctions among extracellular bacterial pathogens, intracellular viral pathogens and intracellular bacterial pathogens and their use of MAC-polyC9, poly-Perforin-1 and poly-Perforin-2. However, the experiments using Perforin-2–/– mice use* S. aureus*, which is primarily an extracellular pathogen although it can invade some cell types. This discussion should be tempered to soften the sharp distinctions that are made.*

The discussion of these three pore-formers has been rewritten. We agree with the reviewer’s statements that the discussion as it was written is confusing and potentially misleading given the findings presented in this study and the accompanying manuscript.

The Discussion talks about 3 immune cell pore-forming proteins (complement, Perforin and Perforin-2). Why is GNLY not mentioned, especially since it has anti-bacterial activity? It would be good to relate the findings here to the recent paper of Walch et al. in Cell.

While we agree that the findings by Walch et al. in Cell regarding GNLY are interesting findings, we feel that GNLY falls outside of the scope of this study. Complement, Perforin, and Perforin-2 are all members of the MACPF super family of pore-forming proteins. Unlike these members, GNLY is a member of the saposin-like protein family. Further, recent manuscripts have demonstrated that while GNLY has anti-bacterial activity when combined with the activity of granzymes, it does not have great lytic activity on its own (requiring micromolar GNLY concentrations or extremely hypotonic or acidic buffers (Ernst, 2000; Stinger, 1998). Due to the above reasons, we do not believe that GNLY would easily mesh with our discussion which focuses on pore forming proteins.